# Human iPS-derived pre-epicardial cells direct cardiomyocyte aggregation expansion and organization in vitro

Jun Jie Tan[1,2,3,13 ✉], Jacques P. Guyette[1,2,13], Kenji Miki [1,2,4], Ling Xiao [2,5], Gurbani Kaur [1], Tong Wu[1,2], Liye Zhu[1,2], Katrina J. Hansen[6], King-Hwa Ling [7,8], David J. Milan[2,9,12] & Harald C. Ott[1,2,10,11 ✉]

Epicardial formation is necessary for normal myocardial morphogenesis. Here, we show that differentiating hiPSC-derived lateral plate mesoderm with BMP4, RA and VEGF (BVR) can generate a premature form of epicardial cells (termed pre-epicardial cells, PECs) expressing *WT1*, *TBX18*, *SEMA3D*, and *SCX* within 7 days. BVR stimulation after Wnt inhibition of LPM demonstrates co-differentiation and spatial organization of PECs and cardiomyocytes (CMs) in a single 2D culture. Co-culture consolidates CMs into dense aggregates, which then form a connected beating syncytium with enhanced contractility and calcium handling; while PECs become more mature with significant upregulation of *UPK1B*, *ITGA4*, and *ALDH1A2* expressions. Our study also demonstrates that PECs secrete IGF2 and stimulate CM proliferation in co-culture. Three-dimensional PEC-CM spheroid co-cultures form outer smooth muscle cell layers on cardiac micro-tissues with organized internal luminal structures. These characteristics suggest PECs could play a key role in enhancing tissue organization within engineered cardiac constructs in vitro.

[1] Center for Regenerative Medicine, Massachusetts General Hospital, Boston, MA, USA. [2] Harvard Medical School, Boston, MA, USA. [3] Advanced Medical and Dental Institute, Universiti Sains Malaysia, Penang, Malaysia. [4] Center for iPS Cell Research and Applications, Kyoto University, Kyoto, Japan. [5] Cardiovascular Research Center, Massachusetts General Hospital, Boston, MA, USA. [6] Worcester Polytechnic Institute, Dept. of Biomedical Engineering, Worcester, MA, USA. [7] Department of Genetics, Harvard Medical School, Boston, MA, USA. [8] Department of Biomedical Sciences, Faculty of Medicine and Health Sciences, Universiti Putra Malaysia, Selangor, Malaysia. [9] Division of Cardiology, Massachusetts General Hospital, Boston, MA, USA. [10] Division of Thoracic Surgery, Department of Surgery, Massachusetts General Hospital, Boston, MA, USA. [11] Harvard Stem Cell Institute, Boston, MA, USA. [12] Leducq Foundation, Boston, MA, USA. [13] These authors contributed equally: Jun Jie Tan, Jacques P. Guyette. ✉email: jjtan@usm.my; hott@mgh.harvard.edu

Directed CM differentiation of human-induced pluripotent stem cells (hiPSCs) employs systematic biochemical treatments to streamline definitive stages of cardiac development, generating spontaneously contracting cardiomyocytes (CMs) at high efficiency in vitro[1–4]. Human iPSC-derived CMs express sarcomeric proteins, exhibit ion channels, propagate cardiac-specific action potentials, and demonstrate excitation-contraction coupling capable of responding to electrical and biochemical stimuli[4,5]. The use of hiPSC-derived CMs may have broad therapeutic applications in cardiac cellular therapy, myocardial tissue engineering, disease modeling, and drug screening for treating the failing heart[6,7]. Recent work showing CM engraftment and function in non-human primate and porcine models of heart failure reinforces the impact of this breakthrough[8,9]. Despite these achievements, hiPSC-derived CMs and resulting cardiac constructs remain phenotypically immature, with underdeveloped organization and electromechanical function[10,11]. Furthermore, bioengineered heart tissues using defined cardiac cells still lack the cellular and structural complexity of native myocardium, which could be greatly improved by implementing techniques that recapitulate and integrate important cues from heart development[12–14].

The biological levers at each stage of directed differentiation hold enormous potential for generating several cardiovascular cell types beyond CMs, which can theoretically be combined to model cardiac development more completely and engineer sophisticated myocardial grafts. By systematically manipulating the onset and duration of biochemical cues at different stages of the cardiac program in vitro, it is possible to shift the fate of hiPSCs to different cardiac cell types[2,15–17]. Studies have shown that the combination of CMs with non-myocyte cardiovascular cells such as fibroblasts, endothelial cells, and smooth muscle cells can enhance morphological and functional maturity[12,13,17–19]. Some of these non-myocyte cell types are known to be the descendants of epicardial-derived cells[20–22].

The epicardium originates from the pro-epicardial organ, a transient organ that emerges from the lateral plate mesoderm (LPM), located proximal to the venous pole of the looping heart during development. The absence of the proepicardial organ (PEO) or epicardium results in underdeveloped ventricles and embryo lethality, due to hindered CM proliferation, myocardial expansion, and coronary vessel formation[23–25]. During heart maturation, epicardial-derived cells integrate within the myocardium and undergo epithelial-mesenchymal transition (EMT) to become fibroblasts, smooth muscle cells, and endothelial cells that enable healthy ventricular thickening, compaction, and angiogenesis[26–28]. Recently, epicardial-like cells have been generated from hiPSCs by modulating the bone morphogenic protein (BMP) and Wnt signaling pathway[15,29,30]. If hiPSC-derived epicardial cells can retain similar effects observed in the developing heart in vivo, they could theoretically be exploited to enhance myocardial complexity and maturity in vitro[28]. However, the effects of these cells on cultured human CMs remain unclear.

In this study, we build upon our previous work in cardiac tissue formation to demonstrate a simple method to generate pre-epicardial cells (PECs) from hiPSCs[4,30], a premature form of epicardial cells expressing typical epicardial genes *WT1, TBX18, SEMA3D,* and *SCX* but capable of developing further to more mature epicardial cells (upregulated additional markers *UPK1B, ITGA4, ALDH1A2*) after being in contact with CMs and recapitulating the developmental roles in embryonic heart formation, including EMT and derivation of fibroblast and smooth muscle cells, and stimulation of ventricular myocyte proliferation partly via RA-dependent IGF signaling of CM proliferation. PEC differentiation is induced via BMP4, retinoic acid (RA), and vascular endothelial growth factor (VEGF) signaling. This method also allows PEC/CM co-differentiation, and the latter reveals spontaneous PEC/CM spatial organization. During indirect co-culture using compartmentalized inserts, PECs show enhanced migration in the presence of CM. Direct co-culture of PECs and CMs in monolayer generates CM networks with improved contractility and efficient calcium handling. In addition, in a facile method to generate and observe interactions in three-dimensional constructs, spheroid co-culture allows for the generation of electrically active cardiac-microtissue constructs with distinct luminal structures.

## Results

**Rapid derivation of PECs from LPM using BMP4, VEGF, and RA.** PECs and CMs have been shown to share a similar pool of cardiac precursors derived from LPM[15,29], and LPM can be derived from hiPSC with prolonged CHIR treatment[8]. We adopted the method and generated LPM progenitors, most of which expressed platelet-derived growth factor receptor-alpha (PDGFRA, 73.7 ± 4.3%) and/or vascular endothelial growth factor receptor-2 (KDR, 88.0 ± 3.6%; Fig. 1ai) at day 3. The 48 h CHIR treatment also demonstrated significant downregulation of pluripotency markers (i.e., *OCT3/4, SOX2, NANOG*), as well as significant upregulation of primitive streak markers (i.e., *MXL* and *T*), LPM marker (i.e., *PITX2* and *FOXF1*), and cardiac progenitor markers (i.e., *GATA4, ISL1,* and *NKX2.5*; Supplementary Fig. 1a and b). BMP4, VEGF, and RA signaling (collectively abbreviated as BVR) are key biochemical factors that drive epicardial differentiation[31]. We induced the LPM with BVR for an additional 96 h, and successfully derived 86.8 ± 4.1% WT1+ cells at day 7 ($n = 3$) within the same monolayer LPM culture without cell splitting or embryoid body formation (Fig. 1aii). WT1+ cells also demonstrated significant increases in *WT1, TBX18, SEMA3D, TBX5, BNC* ($p < 0.0001$), and *SCX* ($p = 0.028$) transcript expression, the known epicardial markers (Fig. 1b). Immunofluorescence staining showed high-level expression of WT1+, TCF21+, and ZO1+ (Fig. 1c), with few cells exhibiting endothelial marker CD31+ (Fig. 1c, Supplementary Fig 1c) or smooth muscle marker SMA+ (Fig. 1c). No cTnT+ cells were detected in any PEC differentiations, despite an upregulation of *TNNT2* gene in PECs. Comparing with differentiating CM, *TNNT2* gene was 235x lower in PECs (Supplementary Fig. 1d). To demonstrate the reproducibility of our differentiation as shown above, this protocol was tested and successfully reproduced using two additional commercial hiPSC lines (Supplementary Fig. 1e, f).

**RNA sequencing revealed PEC profile relatives to known epicardium cells.** Performing an unsupervised hierarchical gene expression clustering analysis, we compared transcriptomic profiles between PECs, expanded PECs at passage 1 (Epi$^{P1}$), and H9-derived epicardial cells (Epi$^{H9}$) from the Palecek group (Fig. 2)[32]. Clustering of the 2,000 most variable genes showed distinct gene expression profiles among PEC, Epi$^{P1}$, and Epi$^{H9}$ cells (Fig. 2a). The Epi$^{P1}$ population (derived from PECs) was clustered closer to the Epi$^{H9}$ population (Pearson's correlation coefficient, $r = 0.88–0.91$) than the PECs ($r = 0.75–0.78$; Fig. 2a). However, the principal component analysis determined that PECs were clustered closer to Epi$^{P1}$ than Epi$^{H9}$ (Fig. 2b). Nonetheless, many of the upregulated DEGs in PECs were found to be downregulated in Epi$^{P1}$ (Fig. 2c). The total DEGs were reduced from 2,696 in the PECs vs. Epi$^{H9}$ comparison (Fig. 2ci) to 903 in Epi$^{P1}$ vs. Epi$^{H9}$-comparison (Fig. 2cii). Of those, 607 of the DEGs were consistently present in both stages. Gene enrichment analysis based on Gene Ontology (GO) for biological processes demonstrated that upregulated DEGs of PECs vs. Epi$^{H9}$ were highly enriched for genes involved in the cell cycle whereas the downregulated DEGs

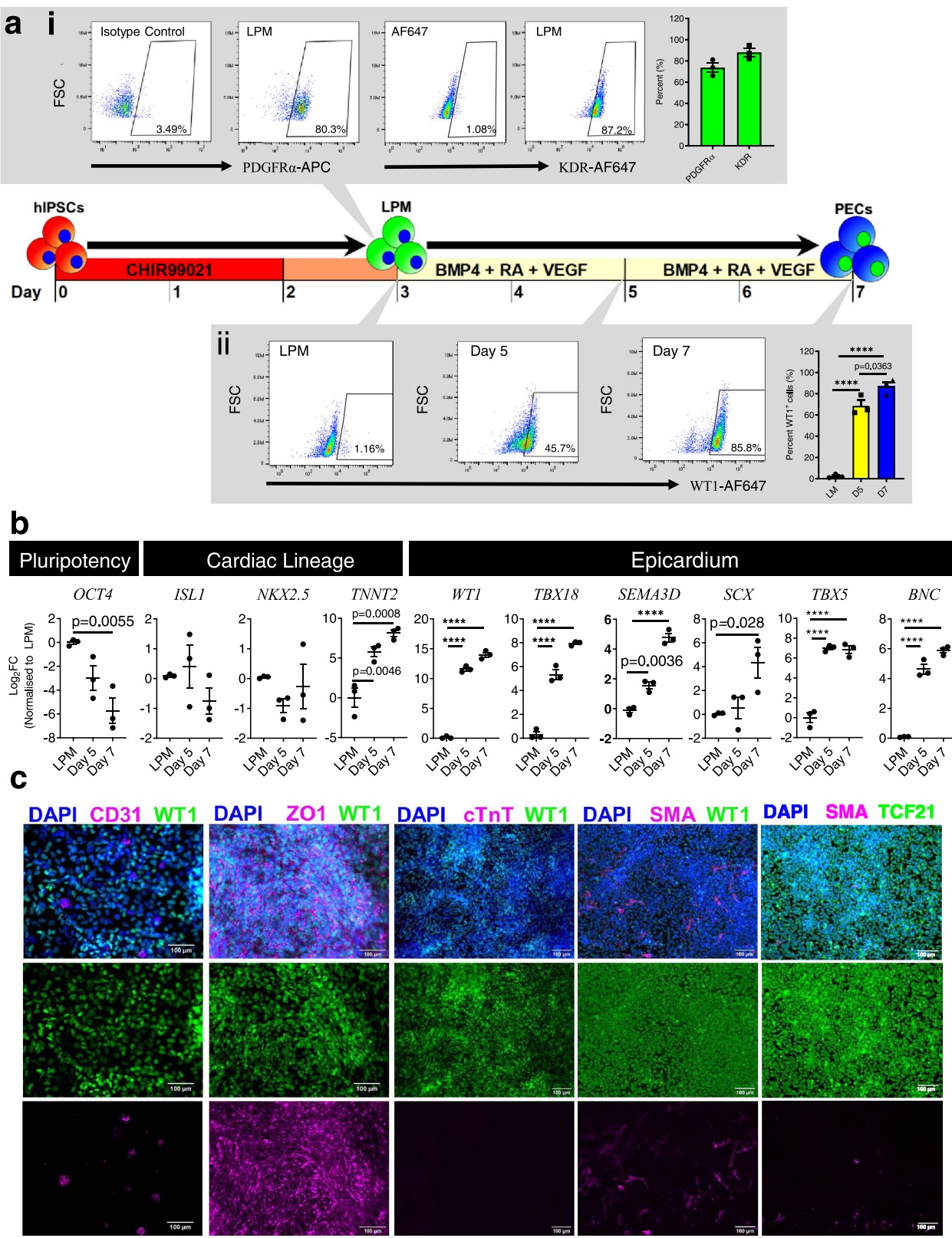

were enriched for angiogenesis, circulatory system development, and cell motility or migration (Fig. 2ci). This is in contrary to Epi^P1 vs. Epi^H9 of which the DEGs involved in the heart, mesenchyme, circulatory and nervous system development were upregulated, while the downregulated DEGs were enriched for

angiogenesis, cell adhesion/regulation, and blood vessel morphogenesis similar to that of PECs vs. Epi^H9 (Fig. 2cii).

To further characterize the differentiation stage of PECs relative to known epicardial lines, the dataset of epicardial transcriptomes from hiPSC-derived (Epi^SC from 19-9-11 and 19-

**Fig. 1 Generation of CMs and PECs from BJ-RiPS cells. a** Schematic shows timeline of CHIR treatment to generate LPM and PECs from hiPSCs. **ai** FACS dotplots show PDGFRA and KDR expressions in LPM at day 3. Bar graph shows mean ± SEM of analyzed from three independent experiments by flow cytometry. **aii** Representative flow cytometric dot plot of WT1 expression in differentiating PECs at day 3 (LPM), day 5 and day 7. Bar graph represents the percent of WT1 at day 3 (LPM), day 5 (yellow) and day 7 (blue) of differentiation. The significant differences between groups and p values were determined by one-way ANOVA with Tukey's multiple comparisons test. *$p < 0.0001$. **b** RTPCR analysis of differentiated PEC at day 3, 5, and 7. Log$_2$ fold change in gene expression was normalized to the level in LPM. Data presented in mean ± standard error of mean (SEM) from three independent experiments ($n = 3$). One-way ANOVA with Dunnett post hoc test was used in the statistical analysis. The presented adjusted $p$ value in all figures corresponds to the group in the respective column versus LPM. ****$p < 0.0001$ versus LPM. **c** Representative immunofluorescence images of day 7 PECs from three independent differentiation demonstrating the expression of WT1, cTnT, CD31, SMA, ZO1, and TCF21. Scale bar = 100 µm.

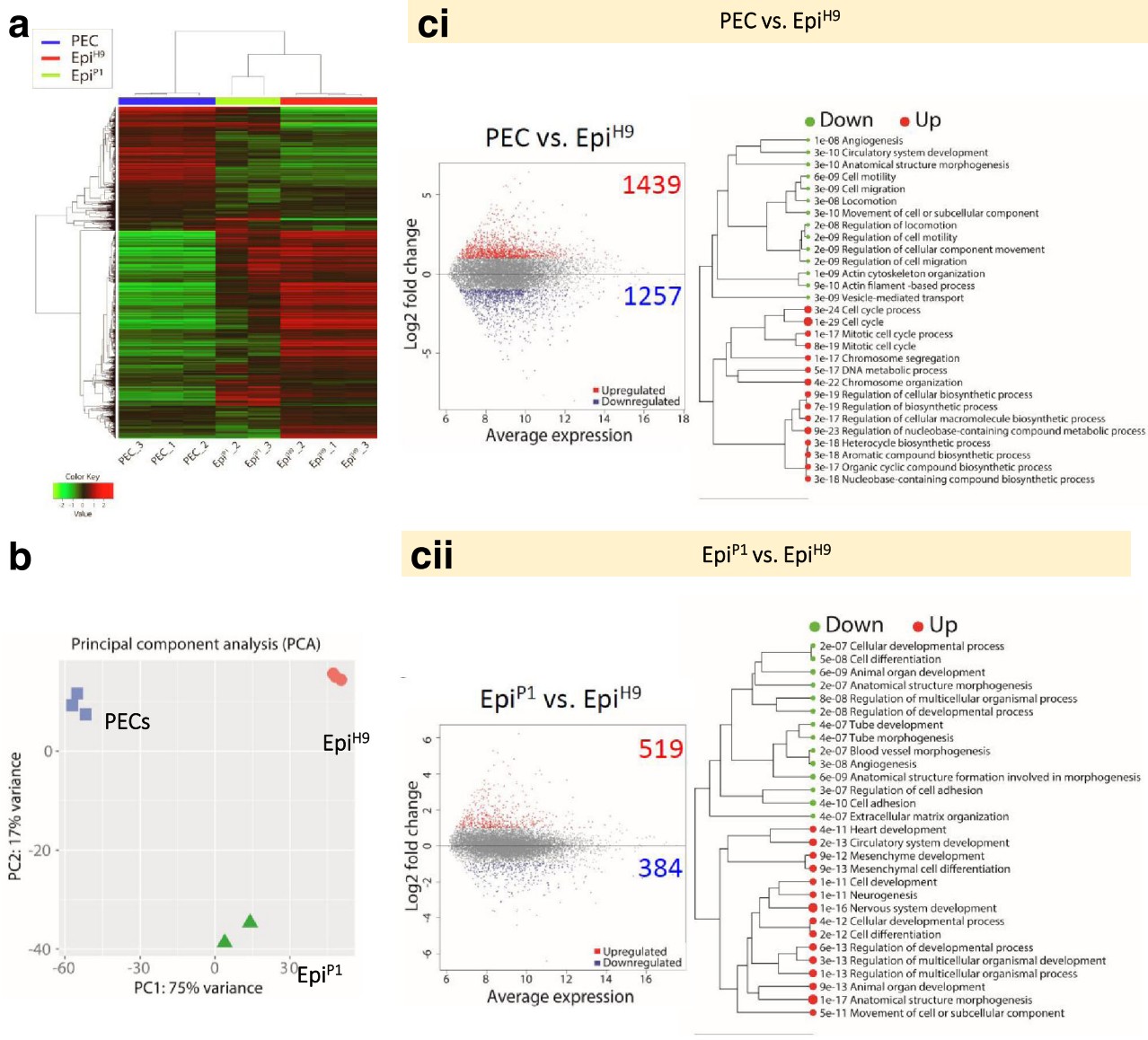

**Fig. 2 RNA Sequencing analysis and functional interpretation. a** Unsupervised hierarchical clustering of 2,000 most variable genes based on correlation distance (average linkage) method. Replicates for each group were from independent differentiation batches. **b** Principal component analysis (PCA) cluster PECs closer to Epi$^{P1}$ as compared to Epi$^{H9}$. **c** MA plots of differential expression analysis for PEC vs Epi$^{H9}$ (i) and Epi$^{P1}$ vs Epi$^{H9}$ (ii) comparisons. Differentially expressed genes (DEGs) are highlighted as red dots for upregulation and blue dots for downregulation (adjusted $p$ value < 0.05 and absolute fold change ≥ 2). Functional clustering of gene ontology terms for biological processes for both up and downregulated DEGs for PEC vs Epi$^{H9}$ and Epi$^{P1}$ vs Epi$^{H9}$ comparisons. DEGs were determined using DeSeq2 and analyzed using the Wald test with Benjamini-Hochberg multiple comparison corrections. FDR value = 0.05 was used as to determine the statistical significance.

9-97 lines), hESC-derived (Epi$^{SC}$ from H9 and ES03 lines), and donor-derived (Epi$^{DONOR}$ from Donor9635, Donor9634, Donor9633, and Donor9605) published by the Palecek group (GSE84085) were included in our analyses[32]. The log$_2$ fold-change of DEGs obtained from PECs vs. Epi$^{H9}$, Epi$^{P1}$ vs. Epi$^{H9}$, and Epi$^{DONOR}$ vs. Epi$^{SC}$ were employed for comparisons to avoid batch effect between the two independent studies. K-means clustering was performed on all the DEGs and functionally group

into seven clusters (A to G) based on enriched gene ontology terms for biological processes (Supplementary Fig. 2a, b). Consistently, the DEGs that were enriched for ontologies related to cell cycle and mitosis (Cluster F) were only found to be upregulated in PECs as compared to Epi[P1] and Epi[DONOR]. Whereas the upregulated DEGs in Epi[P1] are similar to Epi[DONOR], which were enriched for ontologies involving cell differentiation, neurogenesis, circulatory and heart morphogenesis (Cluster A), as well as processes related to cell signaling, metabolism, and communications (Cluster B). Despite the common ontologies, upregulated DEGs from Epi[P1] were uniquely enriched for nervous system development (Cluster G), whereas Epi[DONOR] uniquely enriched for ontologies related to cell motility and angiogenesis (Cluster D).

Then we performed enrichment analysis using the exact hypergeometric probability test on the DEGs lists for PEC and Epi[P1] with the human proepicardial (hProEP) geneset (35 genes) extracted from the 2019 Cui et al. study[33]. The analysis indicated that both PEC and Epi[P1] DEGs were 2.4 ($p < 0.015$) and 4.5 ($p < 0.005$) times more overlaps with the hProEP geneset than expected when compared to the background (28,397 genes), respectively (Supplementary Fig. 2c). When we compare the DEGs to gene set the for epithelial to mesenchymal transition geneset from dbEMT 2.0 previously curated by Zhao et al. (2014, 2019; Supplementary Fig. 2d)[34,35], DEGs from PECs and Epi[P1] were 1.7 ($p < 4.178e^{-14}$) and 2.8 ($p < 1.013e^{-22}$) times more overlapped with the dbEMT 2.0 geneset, respectively. However, DEGs from Epi[DONOR] did not overlap with the geneset significantly.

**Differentiated PECs are capable of epithelial-mesenchymal transition.** PECs were treated with SB431542 (SB) for TBFβ inhibition, to maintain the cuboidal epithelial phenotype with defined ZO1 expression (Fig. 3a) at each cell border in the extended culture. Without TGFβ inhibition, PECs are prone to spontaneous differentiation into SMA+ cells (Supplementary Fig. 1h), a characteristic which has been described in human fetal epicardial cells[36]. Treatment with TGFβ, bFGF, or TGFβ/bFGF for 6 days caused ZO1 disarrangement, cell and nucleus enlargement, and WT1-expression loss in PECs (Fig. 3a). In addition, all three treatments induced new phenotypic expression of mesenchymal marker CD90 (Fig. 3a). The three treatments also significantly downregulated epicardial genes (*WT1* and *TBX18*), E-cadherin gene *CDH1* (except TGFβ/bFGF), and conversely upregulated the expression of N-cadherin gene *CDH2* (Fig. 3bi, ii), suggesting the loss of epithelial identity. Rapid epithelial-mesenchymal transition (EMT) with upregulation *SNAI1* was evident in both bFGF ($p = 0.0404$) and TGFβ/bFGF ($p = 0.0117$) treatments, while *SNAI2* was only increased following TGFβ/bFGF treatment. Activation of bFGF, TGFβ, or both pathways demonstrated the potential of PEC transition towards a mesenchymal lineage, with significant increases in *ACTA2*, *DDR2*, and *POSTN* expression (Fig. 3biii). bFGF appeared to direct PECs toward a fibroblast fate, with significant upregulation of fibroblast markers *TCF21* ($p < 0.0001$) and *VIM* ($p = 0.0358$)[15,32]. Flow cytometric analysis affirmed mesenchymal commitment as bFGF induced 95.3 ± 0.7% of PECs to be CD90+, compared to SB (1.1 ± 0.1%), TGFβ (28.7 ± 3.5%), or TGFβ + bFGF (49.6 ± 2.2%) ($p < 0.0001$) (Fig. 3c). These collective characteristics are akin to epicardial cells[15,32].

**PECs are migratory in the presence of CM, and able to undergo EMT in CM co-culture.** We formulated a "complete PEC medium" containing VEGF, RA, and retinol (abbreviated herein as PECM) for co-culturing PECs and CMs without interfering with TGFβ signaling in the system, which retained WT1+ expression,

maintained PEC phenotype, and minimized spontaneous EMT events (Supplementary Fig. 1g, h). Culturing PECs in PECM alone showed increased TCF21+ and SMA+ cells, along with upregulated *TBX18* ($p = 0.0041$), *ITGA4* ($p < 0.0001$), and *RALDH2* ($p = 0.0004$) in PECs at day 11 (Supplementary Fig. 1i). Compared with stimulation-suppressed PECs by SB, further upregulation in the epicardial-specific marker uroplakin-1B (*UPK1B, p < 0.0001*), as well as epicardial-related genes *TBX18* ($p < 0.0001$), connexin-43 (*GJA1, p < 0.0001*), and α4 integrin (*ITGA4, p = 0.0002$) were observed at days 18 (Supplementary Fig. 1i). Nonetheless, we did not observe upregulation in *RALDH2* expression by day 18, the key epicardial maturity marker indicating RA-production capability.

For initial characterization of PEC-CM interactions in vitro, we co-cultured the cell types in separate compartments of migration inserts and compared against PEC-only seeded inserts. We inhibited PEC/CM proliferation with mitomycin C (MitoC) to negate confounding effects due to cell proliferation. PECs in co-culture became more migratory by 21 h and recolonized a total area of 119.3 ± 21 μm², while PECs-alone only recolonized a total area of 52.3 ± 12.9 μm² ($p = 0.0263$; Supplementary Fig. 3a, b). Time-lapse monitoring over the course of 10 days showed that mCherry-tagged PECs appeared to invade the Venus-tagged CM layer, causing a change in configuration at the PEC-CM border (Supplementary Fig. 3ci, inset, Supplementary Fig. 3d). We observed a layer of mCherry+, SMA+ cells residing in between PECs and sarcomeric α-actinin+ CMs, while those PECs residing in the remote region remained undifferentiated, with a WT1+ SMA- phenotype (Supplementary Fig. 3cvi-x, 3d). A majority of the re-isolated PECs from CM co-culture were found to retain WT1+ and TBX18+ expression (Supplementary Fig. 3e).

**Direct PEC-CM co-culture enables the formation of an integrated network of large CM aggregates.** To further understand the effects of cell-cell interactions during PEC-CM co-culture in vitro, we seeded PECs and CMs together without separation in standard, two-dimensional cell culture. Effects were evaluated against comparators of CMs-alone. We assessed co-cultured PEC fate after 6 days by examining PEC gene expression on FACS-selected mCherry+ PECs from the mixed population. As compared to PEC only culture, RT-PCR results showed significant upregulation of epicardial markers UPK1B ($p = 0.0053$), ITGA4 ($p = 0.0417$) and ALDH1A2 ($p = 0.0003$), as well as retained CDH1 expression ($p = 0.7748$), confirming the transition of PECs to epicardial cells (Fig. 4a); albeit, without changes in GJA ($p = 0.0633$). Consistently, ECAD+, WT1+, and CNN+ cells were also found to present in PEC-CM co-culture, by immunostaining (Fig. 4b).

CMs in co-culture formed dense 3D aggregates by day 8, without discernable cell sloughing or death observed during media changes. The aggregates then formed a connected network by day 16, which became larger and more defined by day 21 (Fig. 4ci). Seeded with the same number of CMs, CM-only controls appeared to have broader and more even coverage across wells, forming fewer and less-pronounced aggregates (Fig. 4cii). Supporting our observations, PEC-CM co-culture significantly condensed CM coverage to 11.79 ± 2.1% of the surface area by day 8, versus 82.48 ± 2.9% coverage in CM-only controls ($p < 0.0001$, Fig. 4ciii), again without a discernable difference in cell death. Networked CM-aggregates in PEC-CM co-culture showed 18.84 ± 2.7% coverage at day 16, which remained significantly <92.65 ± 1.6% coverage in CM-only controls ($p < 0.0001$, Fig. 4ciii). CM aggregates in PEC-CM co-culture formed larger networks by day 21 with 24.07 ± 2.2% coverage but was not statistically different compared to day 16.

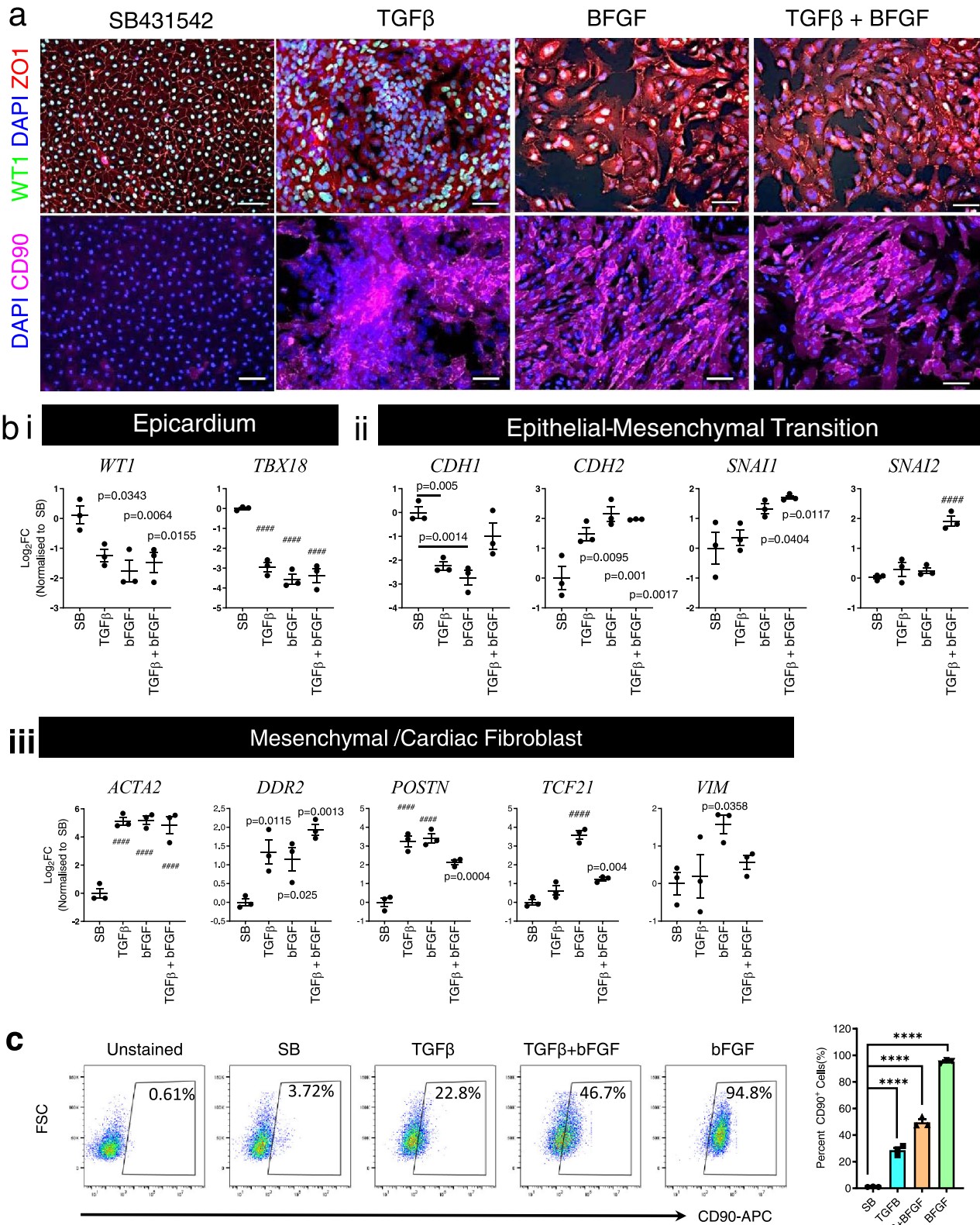

In further characterization of aggregate dimensions, measurements taken by z-plane analysis demonstrated that PEC-CM co-culture aggregates significantly increased in three-dimensional height ($106.1 \pm 4.8$ μm) compared to sparse CM-only aggregates ($79.8 \pm 3.3$ μm, $p = 0.0001$; Fig. 4ciii). Neighboring 3D aggregates in PEC-CM co-culture appeared to form a connected network, capable of synchronized excitation-contraction coupling under both unstimulated and distant electrical point-stimulation conditions (20 V, 0.6 ms, 0.7 Hz; Fig. 4d).

To test if the process could be reproduced when both cell types developed simultaneously, BVR cocktails that drove PEC formation was integrated into CM differentiation protocol to simulate PEC/CM co-differentiation (Fig. 4e). Striking spontaneous PEC/CM spatial organization was observed at day 9 derived from 50.9% of

**Fig. 3 PEC capability of undergoing EMT. a** Representative immunofluorescence characterization of PECs 6 days after treated with SB431542, TGFβ, bFGF, and TGFβ and bFGF from three independent experiments. **b** RTPCR analysis of (i) epicardial gene (ii) epithelial-mesenchymal transition (iii) mesenchymal/fibroblast genes in PECs after 6 days of stimulation. All expressions were normalized to SB-431542 (SB) treated PECs. Data presented in mean ± SEM from three independent experiments ($n = 3$). Statistical significance between group was determined using one-away ANOVA with Dunnett post hoc test. The presented adjusted $p$ value in all figures corresponds to the group in the respective column versus SB-treated PECs. ####$p < 0.0001$ versus SB-treated PECs. **c** FACS analysis of CD90 expression in PECs 6 days after treated with SB431542, TGFβ, bFGF, and TGFβ + bFGF. Overlapping histograms presented in (i), representative FACS dot plots for each group and bar graph with mean ± SEM presented in (ii) ($n = 3$ independent replicates). The significant differences between groups were analyzed using one-way ANOVA with Tukey's post hoc test. ****$p < 0.0001$.

cTnT$^+$ cells and 39.3% of WT1$^+$ cells ($n = 1$). Layers of WT1$^+$ cells were found surrounding stretches of cTnT$^+$ CMs in tube-like structures in the co-differentiation culture (Fig. 4e). Representative video showed contracting vCM after 10 days of co-differentiation (Supplementary Movie 1) and stitched images of a lower magnification showed the overall distribution of cTnT$^+$ CM and WT1$^+$ cells in the co-differentiation culture (Supplementary Fig. 3f, g).

To examine if the observed CM aggregation effects were due to space limitation, PEC-CM co-culture were evaluated against comparators of CMs co-cultured with human umbilical vein endothelial cells (HUVECs) and human cardiac fibroblasts (HCFs). Within 8 days, PEC-CM co-culture appeared to demonstrate an enhanced degree of spatial CM clustering compared to other culture groups, shown by GFP-selected binary image analysis (Fig. 5a). Quantitative assessment at day 8 found that PEC-CM co-culture significantly reduced CM coverage area ($30.4 ± 1.1\%$, $p < 0.0001$ vs. all groups) compared to CM-alone ($78.3 ± 2.8\%$), HUVEC-CMs ($69.0 ± 2.9\%$), and HCF-CMs ($57.6 ± 1.8\%$) indicating PECs-induced CM compaction (Fig. 5a). Heights of three-dimensional aggregates were also measured by z-plane analysis, in which HUVEC-CMs ($34.6 ± 1.3\ \mu m$) and HCF-CMs ($19.4 ± 0.98\ \mu m$) were also significantly less than PEC-CMs ($p < 0.0001$) (Fig. 5b).

**PEC-CM co-culture induces changes in CM electromechanical function.** In addition to morphometric changes, we observed qualitative changes in contractile function under PEC-CM co-culture conditions (Supplementary Movie 2, 3). In quantifying these observations, we again compared PEC-CM co-culture to CMs-alone, HUVEC-CM co-culture, and HCF-CM co-culture. Experimental groups were seeded on 1% collagen gels, and CM contractile strain was analyzed using a high-speed imaging algorithm at 14 days under pacing conditions (20 V, 0.6 ms, 0.7 Hz), a technique we have previously used in several in vitro and in vivo cardiac applications[4,37–41]. CMs in PEC-CM co-culture demonstrated significantly more contractile strain ($3.89 ± 0.3\%$, $p < 0.0001$ vs. all groups) compared to CMs-alone ($1.74 ± 0.33\%$), HUVEC-CM co-culture ($1.08 ± 0.42\%$), and HCF-CM co-culture ($1.12 ± 0.56\%$; Fig. 5c). Post-culture collagen gels were analyzed by atomic force microscopy (AFM) to determine their stiffness. HUVEC-CM seeded gels demonstrated a higher Young's modulus ($1147 ± 86.3$ Pa) compared to CMs-alone ($381.1 ± 32.1$ Pa, $p < 0.0001$), PEC-CMs ($660.6 ± 27.7$ Pa, $p = 0.0162$), and HCF-CMs ($650.4 ± 50.5$ Pa, $p = 0.0001$; Fig. 5d). PEC-CM seeded gels had a significantly higher modulus vs. CMs-alone ($p = 0.0003$), but a similar modulus to HCF-CM seeded gels.

Contractility (force/area) was determined to be significantly enhanced for CMs in PEC-CM co-culture ($0.029 ± 0.002$ mN/mm$^2$) compared to CMs-alone ($0.013 ± 0.001$ mN/mm$^2$, $p = 0.0354$), HUVEC-CM co-culture ($0.012 ± 0.002$ mN/mm$^2$, $p = 0.0123$), and HCF-CM co-culture ($0.008 ± 0.001$ mN/mm$^2$, $p < 0.0001$; Fig. 5e). We hypothesized that differences in contractile mechanics may indicate changes in excitation-contraction

coupling. To characterize differences in calcium handling, we incubated all experimental groups with Fluo-3 AM dye to record [Ca$^{2+}$]i transients at day 14, under electrical field stimulation (20 V, 0.6 ms, 0.7 Hz). [Ca$^{2+}$]i transients were determined within each measured region using spatial averaging of fluo-3AM florescence over time, and then analyzed for amplitude, as well as velocity of calcium transient for both upstroke and decay (Fig. 5f–i). On day 14, CMs in PEC-CM co-culture demonstrated a directionally higher calcium transient amplitude (ΔF/F0: $0.110 ± 0.007$) compared to CMs-alone (ΔF/F0: $0.075 ± 0.009$), but the increase did not meet the threshold for statistical significance ($p = 0.0676$). However, the increased PEC-CM calcium transient amplitude was found to be significantly different than both CM-HUVEC (ΔF/F0: $0.043 ± 0.003$, $p < 0.0001$) and CM-HCF (ΔF/F0: $0.037 ± 0.002$, $p < 0.0001$; Fig. 5f) co-culture groups. Notably, CMs in HUVEC-CM and HCF-CM co-culture demonstrated significantly lower amplitude compared to CMs-alone ($p < 0.0001$). Analyzing calcium transient kinetics, CMs in PEC-CM co-culture demonstrated a significantly increased maximal upstroke velocity (F/F0/sec: $0.0042 ± 0.0003$,) compared to CMs-alone (F/F0/sec: $0.0019 ± 0.0003$, $p = 0.0001$), HUVEC-CM (F/F0/sec: $0.0018 ± 0.0002$, $p < 0.0001$), and HCF-CM (F/F0/sec: $0.0019 ± 0.0002$, $p = 0.0004$; Fig. 5g).

CMs in PEC-CM co-culture also demonstrated a directionally higher but not significantly increased maximal decay velocity (F/F0/sec: $0.00037 ± 0.00004$) compared to CMs-alone (F/F0/sec: $0.00026 ± 0.00003$, $p = 0.2421$). However, the increased PEC-CM maximal decay velocity was found to be significantly higher than both HUVEC-CM (F/F0/sec: $0.00022 ± 0.00002$, $p = 0.0188$) and HCF-CM (F/F0/sec: $0.00014 ± 0.00002$, $p < 0.0001$; Fig. 5h).

In support of the observed PEC-induced improvement in CM contractility and calcium handling, we further assessed the sarcomere length, cell size, and mitochondria content of re-isolated CMs (Fig. 5j–l). The average CM sarcomere length after 6 days of PEC co-culture ($1.82 ± 0.02\ \mu m$; 373 sarcomeres from 16 CMs, from three independent replicates) was found to be significantly longer compared to CMs-alone ($1.72 ± 0.03\ \mu m$, $p = 0.00264$; 230 sarcomeres from 30 CMs, from three independent replicates; Fig. 5j). The percentage of CM with high mitochondria density was also higher after 6 days of PEC co-culture ($79.4 ± 3\%$) vs. CMs-alone ($54.9 ± 6\%$, $p = 0.0029$; Fig. 5k). Nonetheless, there was no significant change in CM cell size after PEC co-culture vs. CM monoculture (Fig. 5l).

**PECs induce CM proliferation partly via the RA-IGF signaling axis.** PECs are involved in stimulating CM proliferation during heart development, with a more prominent effect in the ventricles[24]. To examine this interaction, we first modified our CM differentiation protocol to produce a ventricular myocyte population by inhibiting RA signaling[42,43] using a potent RXR-α antagonist, BMS189453 (BMS), for 96 h starting at day 3 (Fig. 6a). At day 30, CMs in the BMS-treated group exhibited elongated structure (Supplementary Movie 4) and expressed both MLC2A and MLC2V proteins (Fig. 6b), with significantly upregulated ventricular genes MYH7 ($p = 0.0089$) and MYL2 ($p = 0.005$) as compared to control (Fig. 6a, exhibiting a

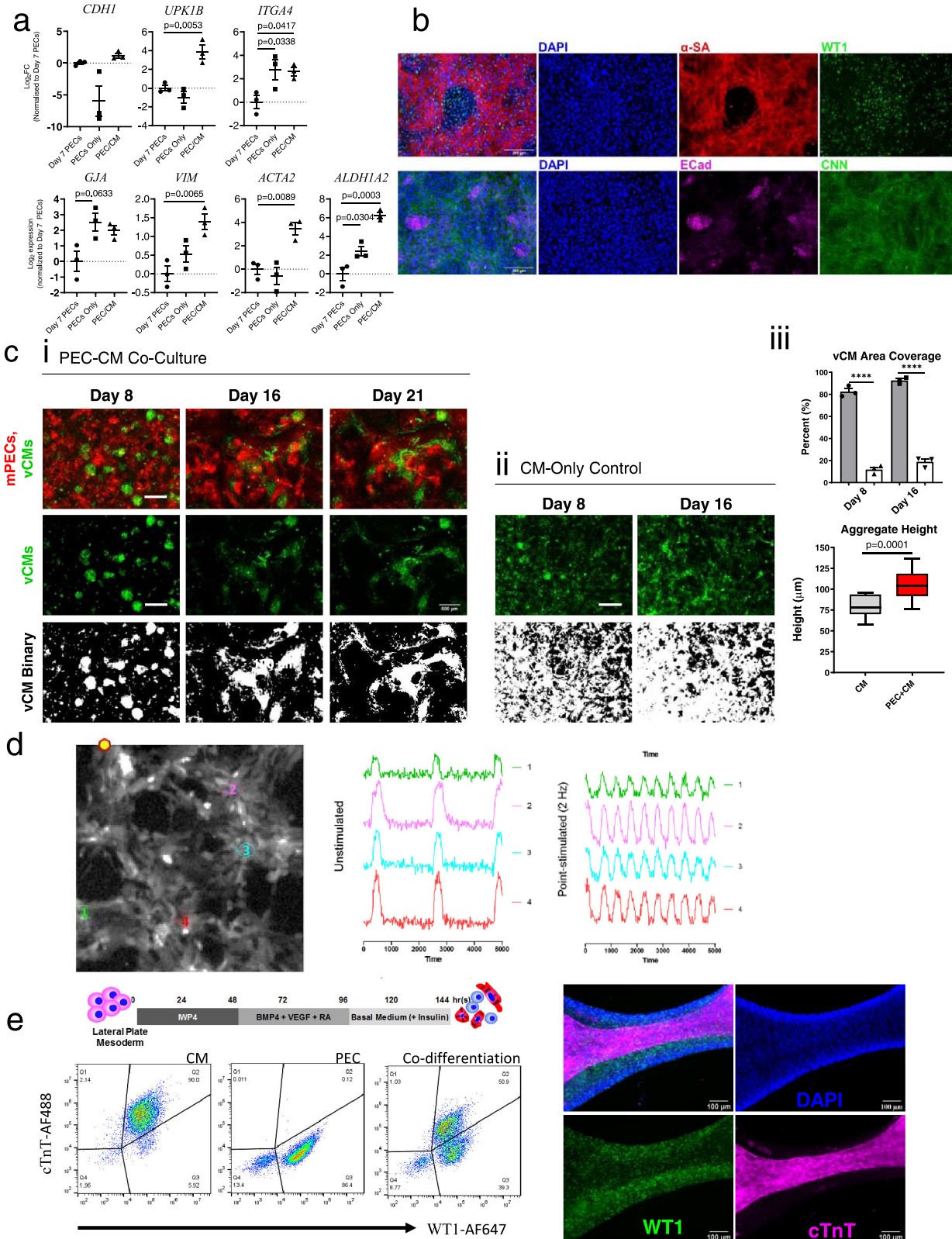

more ventricular-like identity[43]. Consistent with our earlier finding, PEC co-culture caused network formation of CMs after 6 days (Supplementary Movie 5), and cTnT+ EdU+ cells were significantly increased (37.2 ± 2.1%) vs. control CMs without PEC co-culture (28.1 ± 1%, $p = 0.016$; Fig. 6c, d).

IGF signaling plays a key role in PEC-induced ventricular-CM proliferation during early heart development[44]. To test this phenomenon in vitro, ventricular-CMs were treated with Linsitinib (IGF1R inhibitor) for 3 days. Increasing concentrations of Linsitinib gradually and significantly decreased the population

**Fig. 4 PEC fate in CM co-culture and its effect on CM. a** The $\log_2$ fold change of PECs alone and fluorescence-sorted mCherry-tagged PECs from CM co-culture were compared to the expression level in PECs at day 7, analyzed by RT-PCR. Data presented in mean ± SEM from three independent experiments ($n = 3$). Statistical significance versus day 7 PECs was determined using one-away ANOVA with Dunnett post hoc test. **b** Representative images of immunofluorescence-staining from three independent experiments confirm the presence of WT1$^+$ (green) cells (Top) along with α-sarcomeric actinin$^+$ CM (Red) in co-culture. ECAD$^+$ (magenta) epithelial cells and CNN$^+$ (Green) smooth muscle cells were also formed in CM co-culture (bottom). Scale bar = 200 μm. **c** Venus-CM surface area coverage and aggregate formation in (i) PEC-CM co-culture compared to (ii) CM-only controls over time in culture, with (iii) quantification of cell coverage and aggregate height. Scale, 500 μm. Cell coverage measurements were taken from broad-field areas (7.6 mm$^2$) from three independent experiments. Two-tailed student t-test was used for both coverage (bar) and height (box plot) analyses; Data presented in box and whiskers plots represent the maxima, 75th percentile, median, 25th percentile and minima. Aggregate height measurements were taken from $n = 15$ areas from three independent experiments; Data presented as mean ± SEM. ****$p$ values < 0.0001. **d** Representative calcium-signaling traces (using Fluo-3AM) recorded at Day 14 from a broad PEC-CM co-cultured area under (i) unpaced and (ii) paced conditions via point-stimulation (yellow dot; Scale, 100 μm), demonstrating network connectivity at points 1–4, from three independent experiments. **e** Schematic shows timeline for PEC/CM co-differentiation from LPM. Dotplot presents the proportion of WT1$^+$ cells, cTnT$^+$ cells, and double-negative cells in co-differentiation culture ($n = 1$), analyzed by flow cytometry. Immunofluorescence images of co-differentiation culture show representations of spontaneous spatial organization of PECs and CMs in a 'tube' like structure from three independent experiments.

---

of cTnT$^+$ EdU$^+$ CMs in PEC co-culture (21.5 ± 0.8% untreated with 0 nM, to 7.0 ± 0.4% treated with 1 μM, $p < 0.0001$; Fig. 6e), suggesting a significant role of IGF signaling in this process. Furthermore, addition of IGF2 also significantly increased cTnT$^+$ EdU$^+$ CMs in a dose-dependent order (Fig. 6f). Higher IGF2 protein concentrations were detected in PEC-CM co-culture vs. CM-alone after 2 days (983.2 ± 63 pg/ml vs. 325.7 ± 70 pg/ml, $p = 0.0011$) and after 6 days in culture (1979.7 ± 55.1 pg/ml vs. 242.5 pg/ml, $p < 0.0001$; Fig. 6g)[28]. IGF2 level in PECs only culture was similar to CM culture after 2 days (477.9 ± 68.9 pg/ml vs 325.7 ± 69.9 pg/ml) but the level increased to 911.2 ± 16.4 pg/ml/day after 6 days in culture, and the level was significant lower to that of PEC-CM group ($p < 0.0001$).

To study if *IGF2* is expressed in PECs and is stimulated by ventricular-CMs, PECs (250,000 cells/well) were co-cultured with increasing numbers of CMs (0– 500,000 cells/well) in direct co-culture. *IGF2* expression increased significantly in FACS re-isolated mCherry-PECs in response to increasing CM number, in which peak *IGF2* expression occurred during 1:1 cell-type ratio (250k PECs: 250k CMs), eliciting a 2.3-fold increase vs. PEC-alone control ($p < 0.0001$; Fig. 6h). To investigate if *IGF2* expression in PECs is RA-signaling dependent, PEC mRNA was analyzed across three groups in transwell format: (1) PECs-alone, (2) PEC-CM co-culture, and (3) PEC + BMS in CM co-culture. Normalized to PEC-alone culture, the expression of *IGF2* mRNA in co-cultured PECs was upregulated by 107.9-fold. BMS (5 μM) treatment significantly reduced *IGF2* expression in co-cultured PECs vs. control without BMS ($p = 0.0407$; Fig. 6i).

**Co-culture of PEC and CM spheroids enhances the morphogenic complexity within 3D cardiac aggregates.** To investigate if the PEC effects on CM functions observed earlier could be reproduced in three-dimensional (3D) culture and recapitulate PEO-to-heart events during development, we sought to incorporate PECs into differentiated CM-spheres. We first employed a facile technique to generate spontaneously beating CM spheroids from hiPSCs in suspension culture (Supplementary Fig. 4a–d, Supplementary Movie 6), based on previously described techniques. We "inoculated" day 15 CM-sphere spinner flask cultures with day 7 mCherry-PECs (single-cell suspension). Instead of enveloping or integrating with CM-spheres as hypothesized, PECs generated independent spheres after 24 h in spinner flask culture (Supplementary Fig. 5a). Immunostaining 24-h PEC-spheres on cytospin slides showed positivity for WT1 and SMA (Supplementary Fig. 5b). Culturing a sample from the CM/PEC co-sphere suspension in a non-adherent plate under in static conditions, adjacent PEC-spheres and CM-spheres began to

connect and integrate within 24 h (Supplementary Fig. 5c), forming spontaneously beating PEC-CM aggregates and demonstrated excitation-contraction coupling with coupled [Ca2+]i transients and contractile strain (Supplementary Fig 5d, Supplementary Movie 7). Notably, PECs did not demonstrate a calcium transient flux (Supplementary Fig. 5e).

After initial static culture, PEC-CM aggregates could be kept viable under nutation culture with continued beating. The combination of PEC-spheres and CM-spheres into PEC-CM aggregates also appeared to decrease the percentage of TUNEL$^+$ cells (Supplementary Fig. 7a, c). Under continued nutation culture for 15 days, initial histological analysis of the PEC-CM aggregates by H&E showed the formation of organized luminal structures (Supplementary Fig. 7a). The luminal structures in PEC-CM aggregates appeared more organized with dense inner-cell layers than "cysts" that have been previously described in culture and CM differentiation of embryoid-bodies[45–47]. H&E staining of our CM-spheres also demonstrated the loosely unorganized cystic structures previously described. In-depth characterization of individual PEC- and CM-spheres during its formation, differentiation, and co-culture are described in supplemental information (Supplementary Fig. 6a–h).

Morphologic analysis of day 15 PEC-CM aggregates (formed with mCherry-PEC-spheres and Venus-CM-spheres) allowed for localization of key features within the morphogenic structures. Based on the distribution of mCherry and Venus-GFP signal, PECs appeared to be integrated with CMs (Supplementary Fig. 7d). GFP$^+$ signal of CMs coincided with a positive sarcomeric α-actinin signal. Undifferentiated PECs demonstrated WT1$^+$ expression. Some differentiated PECs appeared to lose WT1 expression but maintained an mCherry$^+$ signal, which also co-localized with VE-cadherin$^+$ (endothelial) and SMA$^+$ (smooth muscle) expression (Supplementary Fig. 7d). Luminal structures appeared to be highly organized with densely ordered cells lining the lumen regardless of size (Supplementary Fig. d inset) vs. cyst-like structures previously described in embryoid body (EB) studies[45–47]. Histological analysis revealed there was variation in structure makeup. Some lumens were lined with VE-cadherin$^+$ cells (Supplementary Fig. 7d, blue arrows), as previously seen when EBs are preconditioned with shear stress[48]. Yet, other luminal structures were lined with VE-cadherin$^+$ cells surrounded by an SMA$^+$ cell layer (Supplementary Fig. 7d, gold arrows). In a whole-mount staining of PEC-CM aggregates after 15 days in co-culture, z-stack analysis confirmed the formation of SMA$^+$ cell layer to the adjacent cell aggregates (Fig. 7a). This outer SMA$^+$ cell layer was not observed in CM only aggregates which were positive for MLC2V (Fig. 7b).

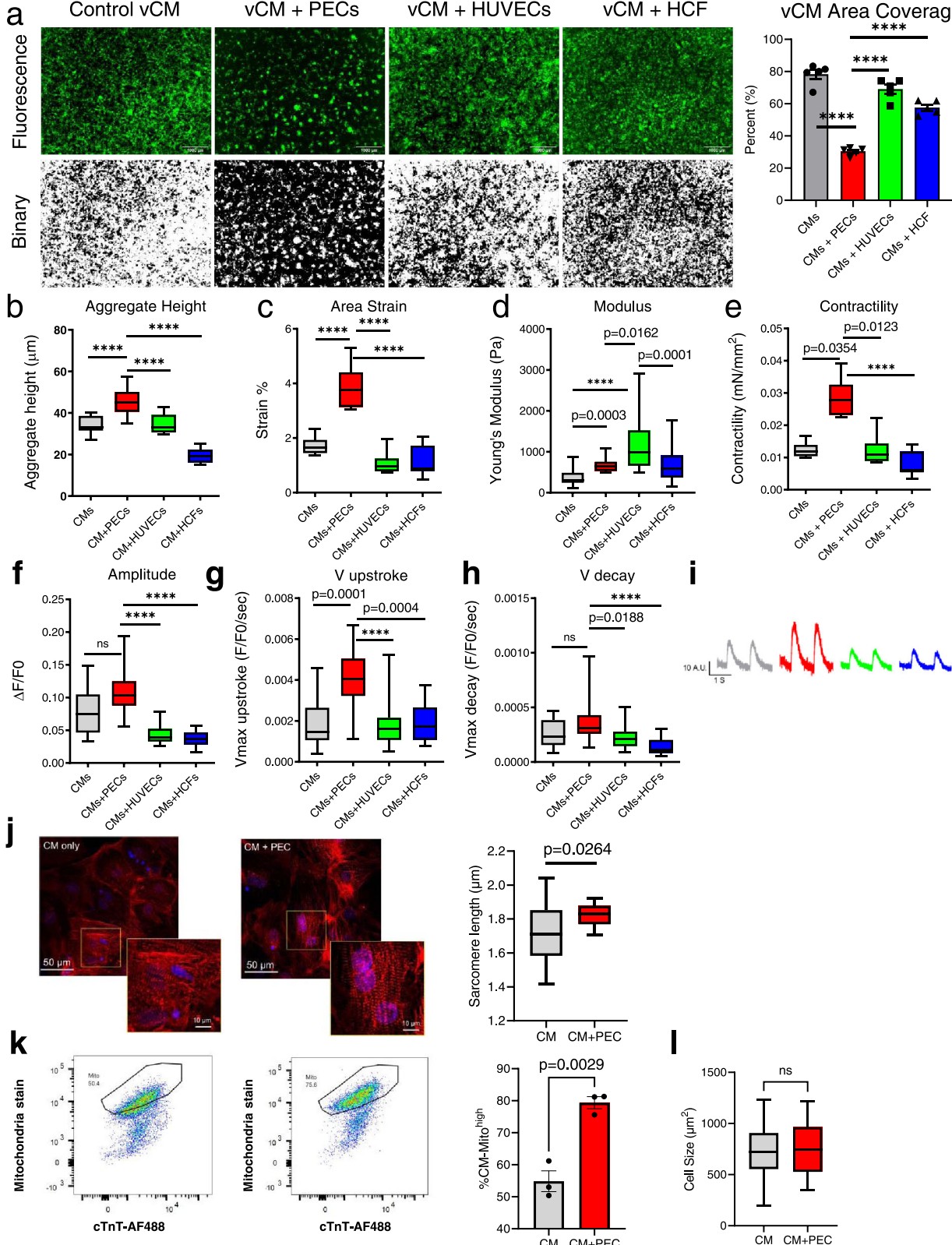

## Discussion

The PEO is a transient embryonic organ that envelopes the developing heart and eventually forms the epicardium. It drives several developmental events that contribute to cardiac architecture and function including CM proliferation, coronary vessel extension, and myocardial compaction. These capabilities suggest that PEO-derived cells, or the transitioning, intermediate cells prior to forming the epicardial cells (the PECs), play key roles in myocardial regeneration, which are likely critical factors that would facilitate the creation of thick myocardial tissues in vitro. Recent reports of successful derivation of epicardial cells from hiPSCs marks a significant milestone, enabling exploration of their potential at earlier primitive stages such as the pre-epicardium[15,29].

**Fig. 5 Effects of PECs on 2D CM co-culture. a** Representative fluorescence and binary images from five independent replicates of vCMs alone and co-cultured with either HUVECs, human cardiac fibroblasts (HCFs), or PECs after 8 days. (Right) vCM surface area coverage in co-culture with HUVECs, HCFs, and PECs compared to CM-only controls after 8 days. Scale, 500 μm. Cell coverage measurements at 8 days were taken from broad-field areas (7.6 mm$^2$, $N = 5$ for each group); Data presented as mean ± SEM and analyzed by using one-way ANOVA with Dunnett post hoc test versus PEC/CMs only. ***$p$ value < 0.0001. **b** Two-dimensional aggregate height analysis of all CM culture groups (CMs-alone, PEC-CMs, HUVEC-CMs, HCF-CMs) at day 8 of culture. $N = 15$ aggregates examined over three independent experiments. Data are presented in box and whiskers plot, with lines indicate 25th, 50th, and 75th percentiles and minima and maxima. One-way ANOVA with Dunnett post hoc test versus PEC/CMs was used. ****$p$ value < 0.0001. **c** Cardiomyocyte contractility metrics. Area strain of cardiomyocytes from different experimental groups, determined using high-speed HDM imaging analysis, with $n = 7$ areas per CM group, three contractions analyzed per area examined over three independent replicates. Data are presented in box and whiskers plot, with box lines indicating 25th, 50th, and 75th percentiles, and whisker lines indicating minima and maxima. One-way ANOVA with Dunnett post hoc test versus PEC/CMs was used. ****$p$ value < 0.0001. **d** Young's modulus of cell-seeded gels from different experimental groups, determined by atomic force microscopy. Data were analyzed from $n = 41, 29, 48,$ and $47$ areas of CMs only, CM + PECs, CM + HUVECs and CMs + HCF, respectively examined over three independent replicates. **e** Day 14 contractility (force/area) of CMs cultured alone or in co-culture with PECs, HUVECs, or HCFs (n=7). **f-i** Day 14 Fluo-3AM calcium-signaling analysis of CMs cultured alone or in co-culture with PECs, HUVECs, or HCFs for amplitude (**f**), maximal velocity of calcium transient upstroke (**g**), and maximal velocity of calcium transient decay (**h**). Representative calcium traces are depicted for comparison (**i**). ($n = 15$ for each group, across all calcium transient metrics); **d–i** Data are presented in box and whiskers plot, with box lines indicating 25th, 50th, and 75th percentiles, and whisker lines indicating minima and maxima. Statistical significance at $p < 0.05$ was determined using Kruskal–Wallis with Dunn's multiple comparison tests. ****$p$ value < 0.0001. **j** Representative images of α-Sarcomeric actinin-stained CM in and PEC co-culture (red = α-actinin, blue = DAPI). Sarcomere length was measured manually from a total of 30 sarcomeres of three control CMs and 30 sarcomeres of three PEC co-cultured CMs derived from three independent biological replicates. The line presented in the boxplot is 25th, 50th, and 75th percentiles and minima and maxima. Comparison between CM alone vs PEC/CM were analyzed using a two-tailed *T*-test with Welch's correction. **k** Representative FACS analysis of mitochondria stained CMs in control (left) and PEC co-culture (right). Bar graph showed the percent of CMs with high mitochondria density (% CM-Mito$^{high}$). A two-tailed student *t*-test was used ($n = 3, p < 0.002$). **l** CM size analysis comparing CM ($n = 129$) vs PEC/CM ($n = 115$). The line presented in the boxplot is 25th, 50th, and 75th percentiles and standard deviation. Comparison between CM alone vs PEC/CM were analyzed using a two-tailed *T*-test with Welch's correction.

Studies have shown that BMP4, RA, and Wnt signaling were key mechanisms driving PDGRFA$^+$ and KDR$^+$ LPM cells or cardiac progenitors, to a epicardial fate[15,32], some of which observed better efficiency with simultaneous activation of the signaling[29,31,49]. In this work, we adapted the signaling model to develop an efficient BMP4/VEGF/RA based protocol, which generated >86% WT1$^+$ cells in 7 days from monolayer hiPSC culture without the need of embryoid body generation, or fluorescence sorting. Differentiated PECs demonstrated ZO1, TBX18 gene expression, as well as significant upregulation of the transcription factor *TBX5* that is critical in proepicardium development and specification[50,51]. PECs also shared similar morphological descriptions and functional characteristics as previously reported, including a cuboidal phenotype after in vitro passaging and capability to undergo EMT[15].

Based on the evidence that activated adult epicardial cells were heterogenous and molecularly different to embryonic epicardium[52], and previous studies showed a consistent smooth muscle fate in the administered mature iPSC-derived epicardial cells in vivo, we speculated that the formation and maturation of epicardium occurs only after being in contact with the myocardium. Hence, this study focused more on the use and functionality of the PECs. We define PEC as the premature form of epicardial cells prior to the exposure to signals deriving from the CMs, or the niche which promotes further cell development into forming more mature epicardial epithelium and the descendant epicardial-derived cells. PECs were able to mature in CM co-culture with significant upregulation of *RALDH2* and several mature epicardial genes. We used Epi$^{H9}$ from Palecek's lab in our experiment as an alternative comparator to bona fide epicardial cells and analyzed against the published dataset. We employed log$_2$ fold-change of DEGs obtained from PEC vs Epi$^{H9}$, Epi$^{P1}$ vs Epi$^{H9}$ and Epi$^{DONOR}$ vs Epi$^{SC}$ comparisons to reduce the variations between independent studies and found that the DEGs of Epi$^{P1}$ correlated with Epi$^{DONOR}$, the adult human epicardial cells. Gene ontology analysis also suggests that the DEGs of Epi$^{P1}$ and Epi$^{DONOR}$ are both enriched for heart and circulatory system development. Collectively, these data confirm the identity of PECs and their potential to become more mature epicardial cells.

To better describe the stage of development of the PEC population used in this study, we referred to the most recent publication which delineates the molecular signatures in proepicardial cells in human fetal hearts[33], a cell population which often refers to the derivation from PEO capable of migrating to the heart to form epicardium. Comparing the published gene set from human samples, we found that *HEY1*, one of the putative markers of proepicardium was upregulated in PECs as compared to Epi$^{H9}$; whereas *C1S*, the complement component predominantly expressed in mature epicardial cells, was downregulated in PECs (Fig. S2c). Furthermore, we also observed high cTnT$^-$ EdU$^+$ cells in CM co-culture (Fig. 6e–f), which coincides with the GO functional analysis indicating that PECs are more proliferative than the Epi$^{H9}$ (Fig. 3cii). These results also coincide with the observation found by Cui et al. that proepicardial cells are the most actively cycling cells among all non-immune cells in human fetal heart. Despite the molecular and functional similarity that suggest the resemblance of PEC to proepicardium, the caveat of this study is the lack of in vivo evidence to distinguish between the identity of proepicardial cells and the mature epicardial cells. Hence, we maintain the term to define our cells as PECs.

The morphologic and functional impact of PECs during development lies in their interactions with the myocardium. Successful application of hiPSC-derived PECs ex vivo or in vivo may require achievement of definitive milestones: formation and maintenance of self-renewing epicardial epithelial layer, derivation of epicardial-derived cells capable of EMT, and activation of biochemical signaling responsible for myocardial developmental features (e.g., proliferation via RA signaling)[53]. We are first to demonstrate two of the most fundamental functions that human PECs are capable of, which have not been shown elsewhere using mature human epicardial cells: (1) the ability to form epicardial epithelium, and (2) the ability to express and secrete IGF2 in PEC/CM co-culture, recapitulating aspects of in vivo development. The expression of IGF2 in PEC-derived cells had also increased the number cycling CMs in PEC/CM co-culture. Further elucidation of the downstream signaling mechanism underlying PEC/CM interactions, such as ERK[54], or Yap/Taz

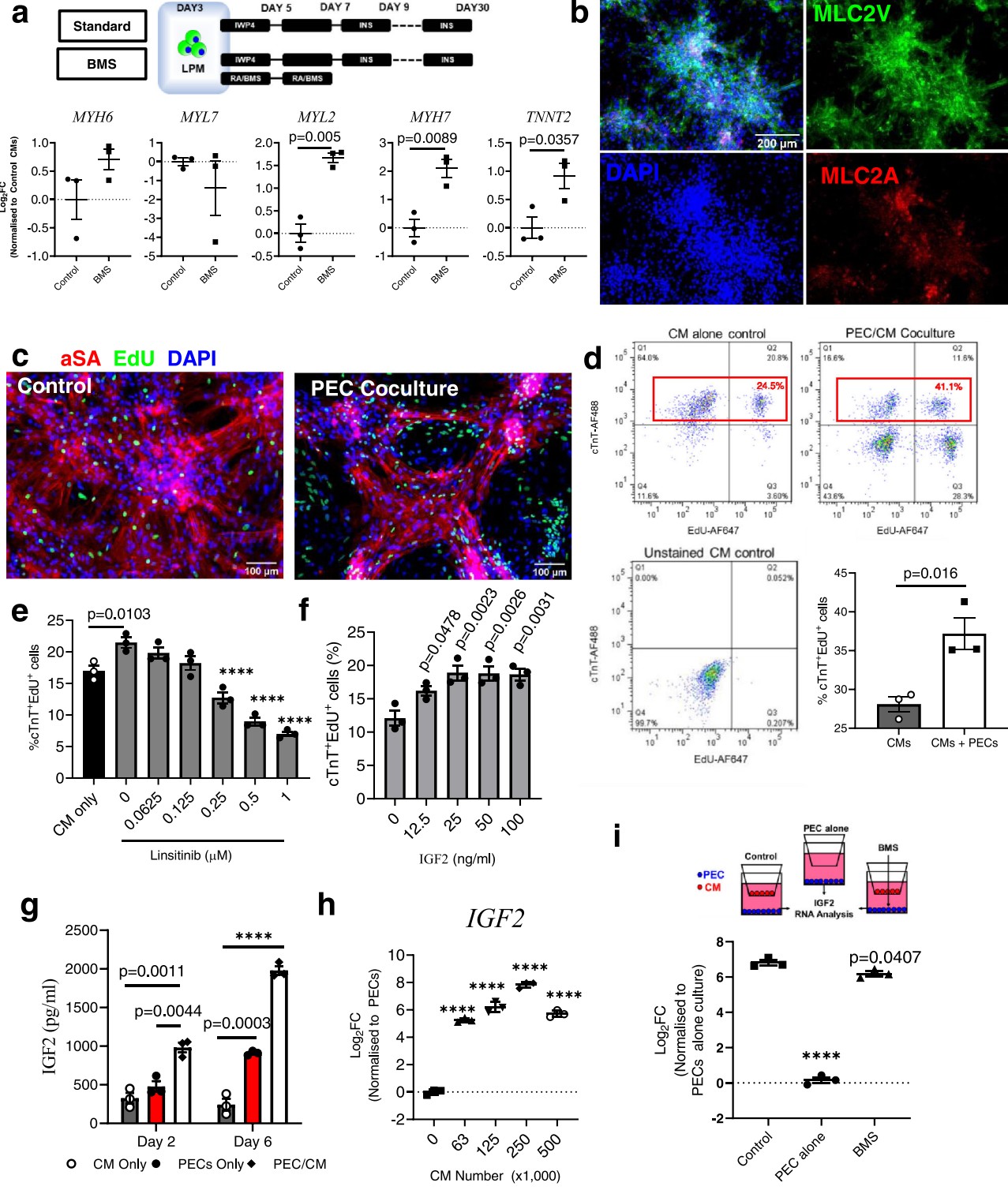

Hippo pathway[55] would warrant greater control in recapitulating PEC-mediated heart development in vitro.

In well-insert experiments, PECs migrated towards CMs to close the gap within 12-21 h (Fig. S3a), and appeared to form an PEC-layer at 24 h which persisted over 10 days (Supplementary Fig. 3c). Notably, PECs did not overtake the CM culture, but did appear to consolidate CMs into a more confined area. Direct co-culture of PECs with CMs appeared to consolidate CMs into dense aggregates, significantly reducing CM coverage compared to CMs-alone, HUVEC-CM co-culture, or HCF-CM co-culture within 8 days (Fig. 5a). Previous work in developmental biology has shown that epicardial cells and signaling are necessary for thick ventricular-wall formation through mechanisms of myocardial growth and compaction[23,24]. It is likely that PEC contact or signaling facilitated CM organization into thicker, networked formations (Fig. 4c). Integrating BVR into CM differentiation showed successful derivation of both PEC/CM in single hiPSC culture, and spontaneous PEC/CM spatial organization (Fig. 4e).

**Fig. 6 Differentiation of ventricular-committing CM (VM) and IGF-RA Signaling in PEC induced VM proliferation. a** Schematics timeline and protocol of ventricular-like CM differentiation from lateral plate mesoderm (LPM, Top), and its cardiac gene expression at day 30 (Bottom) from three independent differentiations. Data presented as mean ± SEM. Unpaired ttest with Welch's correction was used for statistical comparison between groups. **b** Representative immunophenotypic characterization of ventricular-CM at day 30 from three independent differentiation. BMS = retinoid-x-receptor antagonist BMS189453. **c** Representative immunofluorescence images of Sarcomeric Actinin (Red) EdU$^+$ (green) ventricular CMs after co-culturing with PECs from three independent experiments. **d** Representative flow cytometric quantification of cTnT$^+$ Edu$^+$ ventricular CM in PEC co-culture after 6 days. The percentage in red was calculated by taking the cell count in Q2 and dividing by 100% cTnT$^+$ cells, or total events in Q1 and Q2 only (red box). Data presented in mean ± SEM ($n = 3$). Two-tailed student $T$ test was used. **e** Percentage of cTnT$^+$ EdU$^+$ VM in response to Linsitinib. Data presented in mean ± SEM ($n = 3$ independent experiments) versus 0 µM Linsitinib. One-way ANOVA with Dunnett post hoc test was used. **f** Dose-response examination of IGF2 on CM proliferation. Data presented in mean ± SEM ($n = 3$ independent experiments). One-way ANOVA with Dunnett post hoc test was used. **g** ELISA analysis of IGF2 in conditioned medium 2 and 6 days after co-culturing with PECs. Data presented in mean ± SEM ($n = 3$ independent experiments). One-way ANOVA with Dunnett post hoc test was used. ****$p$ value < 0.0001 versus CM only. **h** The expression of *IGF2* RNA in PECs in response to increasing CM number. Data presented in mean ± SEM ($n = 3$ independent experiments). One-way ANOVA with Dunnett post hoc test was used. ****$p$ value < 0.0001 versus no CMs. **i** *IGF2* expression in PECs in CM co-culture in transwell with or without RARα/γ antagonist BMS-189453 (BMS). Data presented in mean ± SEM ($n = 3$ independent experiments). One-way ANOVA with Dunnett post hoc test was used. ****$p$ < 0.0001 versus PECs-CM co-culture without BMS.

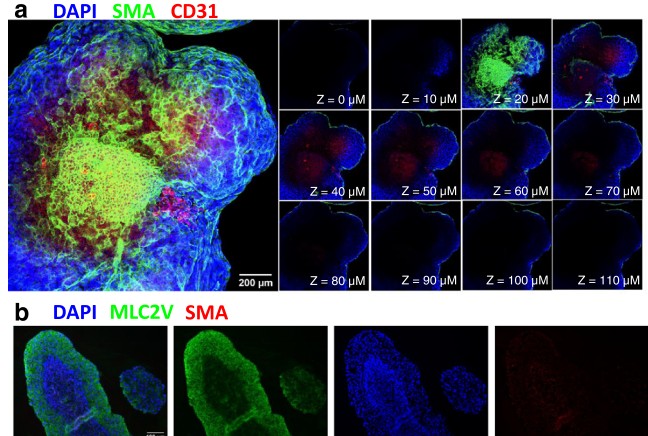

**Fig. 7 PEC organization on CM aggregates. a** z-stack images of the whole-mount PEC-CM aggregates were fluorescent stained for SMA (green), CD31(red) and DAPI (blue) of after 15 days ($n = 1$). Scale = 200 µm. **b** Immunostaining of day 15 CM only aggregates for MLC2V (green), SMA (red) and DAPI (blue). $n = 3$ independent experiments, Scale bar = 100 µm.

Consolidated CM arrangement may also partially account for the observed differences in mechanical function, as densely aggregated CMs provide more contractile units/area working together as a syncytium (Fig. 5a). Under pacing conditions, CMs co-cultured with PECs demonstrated nearly 3x as much contractility (force/area; $0.029 ± 0.002$ mN/mm$^2$) than CMs-alone, HUVEC-CM co-culture, or HCF-CM co-culture within 7 days (Fig. 5e). In two-dimensional culture, the PEC-CM combination generated contractility on the order of engineered myocardial tissue[56]. Beyond the suggestion that the mechanical differences are solely due to condensed contractile CMs, it may also be possible that PECs may have induced CM-proliferation to add more contractile units, as evidenced by our results in Fig. 6c, d. As CMs in PEC co-culture demonstrated longer, more mature sarcomeres and increase mitochondria content, it is also possible that PEC-CM co-culture could have facilitated increased contractility through either improved length-dependent CM activation or enhanced CM maturity (Fig. 5j, k). Nonetheless, the CM size did not significantly differ in the presence of PECs. This observation could be a result of increased small CM population due to proliferation or the short co-culture time period (6 days). In Bargehr's study, co-culture of epicardial cells enhanced CM cells after 14 days[57]. More work is needed to understand the possible effects of PEC-CM cell interactions on CM morphology, sarcomere component isoform switching, mitochondrial structure/function, metabolic bioenergetics, channel, and regulator expression, and transverse tubule formation.

In evaluation of calcium handling, CMs in PEC-CM co-culture demonstrated significantly increased [Ca$^{2+}$]i transient amplitudes compared to other co-culture groups, as well as directional improvement vs. CMs-alone (Fig. 5f). In analyzing [Ca$^{2+}$]i transient kinetics, PEC-co-cultured CMs also showed significantly increased maximal upstroke vs. all other groups (Fig. 5g); while decay velocity was significantly increased compared to HUVEC-CMs and HCF-CMs, and directionally improved vs. CMs-alone (Fig. 5h). Taken together, these results suggest that CMs co-cultured with PECs may have more efficient calcium handling, especially in consideration of the enhanced mechanical contractility. Increased calcium transient amplitudes observed in PEC-CM co-culture could suggest that PEC-induced CMs have increased Ca$^{2+}$ release by sarcoplasmic reticulum—a mechanism that could support enhanced inotropy. The mechanisms behind this augmented excitation-contraction coupling begs further investigation with respect to the expression and function of key calcium channel proteins such as sarco/endoplasmic reticulum calcium-ATPase 2 (SERCA2), L-type calcium channel (LTCC), and Na$^+$/Ca$^{2+}$ exchanger (NCX), ryanodine receptor 2 (RYR2), and phospholamban[56]. In addition, potential evidence for transverse tubule organization and junctophilin-2 expression would likely suggest CM functional maturation[58]. Further, investigating the expression of atrial markers (e.g., MLC1a, sln, Gja5) and ventricular markers (e.g., MLC1v, phospholamban) will help to determine if CM selection plays a role in electromechanical outputs[42]. The effects of PEC-CM co-culture on CM electrical maturation are not clear. Future electrophysiological studies on isolated CMs from PEC-CM co-culture will be necessary to support the enhanced CM electromechanical function in the presence of PEC.

In previous PEC studies, treatment with TGFβ-inhibitor SB431542 stimulated PEC proliferation and inhibited spontaneous EMT in vitro, whereas uninhibited PECs spontaneously upregulated SMA expression[32]. In consideration of the complexity of cardiac biochemical- and mechanical-signaling networks within the heart, it remains unclear why hiPSC-derived epicardial cells transitioned predominantly to a smooth muscle fate after in vivo intramyocardial injection[32,59]. Little data supports the notion that injected PECs could retain their identity as epicardial cells without transplantation techniques that allow for epicardial-surface interaction with myocardium. We speculate that transplantation of PECs would require strategies that take PEC-CM spatial organization into account, to rebuild the

functionally active "embryonic" epicardium. In a simplified in vitro model of three-dimensional tissue, we generated CM-spheres and PEC-spheres, and then combined them to for PEC-CM aggregates. After 15 days of sphere co-culture, H&E staining of the PEC-CM aggregates showed well-organized phenotypically complex luminal structures (Supplementary Fig. 7) that appeared to be distinct from cystic structures previously described in embryoid-body studies[45–47], and more indicative of recent evidence of high-order architecture and cellularity[48,60,61].

In day 15 PEC-sphere and CM-sphere controls, PEC-spheres also contained several well-organized luminal structures, while CM-spheres generated typical cystic structures. Analyzing the development of these structures days −7 to −1 prior to sphere co-culture (Supplementary Fig. 6), the immediate presentation and persistence of CM-sphere cysts would suggest these could result from culture conditions (e.g., spin-culture speed, diffusion/necrosis)[48,62]. However, the gradual development of more complex PEC-sphere structures would suggest more sophisticated mechanical (e.g., spin-culture) and biochemical (e.g., EMT) interactions. Through serial-section immunostaining of PEC-CM aggregates, we observed PEC-derived cells that had undergone EMT to generate SMA+ or VE-cadherin+ cells in the luminal structures (Supplementary Fig. 7d). We did not examine if the VE-Cadherin+ cells were descended from PECs. Hence it is unknown if the VE-Cadherin+ cells were differentiated from PECs, or a byproduct from the PEC differentiation culture.

These results coincide with additional evidence that PECs can undergo EMT to demonstrate SMA, vimentin, periostin and CD90 markers when co-cultured with CMs in 2D culture (Fig. 2b, c). Lessons from cardiac development suggest these luminal cavities could be early indications of vascular structures[63], but their purpose, development, and functionality remain unclear. The observation that VE-cadherin+ endothelial-like cells and E-cadherin+ epithelial-like cells can integrate within luminal structures could indicate that cell-cell communication plays a key role in formation, order, or purpose of the structures (Supplementary Fig. 7d). Alternatively, it also is possible that functions of the luminal structures could influence surrounding cell phenotypes. It is likely that tunable mechanical and biochemical mechanisms can be adjusted to alter the generation or development of these structures, but we did not test these parameters in this study.

One of the most important aspects of PECs are their role in ventricular expansion during heart development[53]. Similar to previous studies, we also found that inhibition of retinoic acid signaling can promote ventricular specification[43,64]. We recapitulated the PEC-CM relationship through in vitro co-culture to show that PECs induce CM proliferation via partial dependence on IGF signaling[44]. In 2019, Bargehr et al.[57] revealed that co-transplantation of CMs with embryonic stem cell-derived epicardial cells increased CM proliferation, contractile strength, and Ca$^{2+}$ dynamics. They also witnessed smooth muscle encoatment of host vessels by the transplanted epicardial cells in the chorionic vasculature of chick embryos. These findings are also in line with our observations presented in this study, suggesting that our differentiated PECs are also likely capable of recapitulating these developmental events in vitro. To understand these phenomena more completely, continued study is required to elucidate underlying mechanisms.

Taken together, this evidence suggests that hiPSC-derived PECs are functional and able to interact with CMs to enhance the function and structural organization in three-dimensional PEC/CM microtissues. We believe this work lends to continued efforts in directed morphogenesis, in which the strategic combination of early-stage cardiac cell types can enable the creation of more sophisticated and mature cardiac grafts.

## Methods

**HiPSC culture, maintenance, and differentiation.** Human BJR-iPS cells (hiPSCs) were obtained from Harvard Stem Cell Institute, and maintained in MTeSR medium (STEMCELL Technologies). hiPSCs were passaged with ReleaSR (STEMCELL Technologies, Cambridge, MA) and plated at 1:20 ratio every 7 days onto 10 cm$^2$ dish pre-coated with growth factor reduced (GFR)-Matrigel (Corning, Tewksbury, MA). Briefly, hiPSCs were seeded at 200,000 cells/cm$^2$ onto a GFR-Matrigel-coated plate and maintained in MTeSR medium for 4–5 days. To generate LPM, were treated with 12 μM Stemolecule$^{TM}$ CHIR99021 (Stemgent, San Diego, CA) in differentiation basal medium consisted of RPMI medium supplemented with B27 without insulin (Gibco, Grand Island, NY) (RPMI-INS,) for 48 h. Then, the medium was replaced with RPMI-INS for 24 h. Next, the culture was incubated with 5 μM of Stemolecule$^{TM}$ Wnt inhibitor IWP-4 (Stemgent, San Diego, CA) in RPMI-INS for 48 h before switching it back to RPMI-INS, for an additional 48 h. After that, the medium was changed to RPMI medium supplemented with insulin-containing B27, and refreshed every 2 days for long-term maintenance. For PEC differentiation, LPM cells at day 3 were treated with RPMI-INS supplemented 50 ng/ml bone morphogenic protein (BMP) 4 (PeproTech, Rocky Hill, NJ), 5 ng/ml vascular endothelial growth factor (VEGF; PeproTech, Rocky Hill, NJ) and 4 μM retinoic acid (RA; Stemgent, San Diego, CA) in RPMI-INS and changed every 2 days for 96 h. PECs were dissociated using collagenase I (Sigma Aldrich, St. Louis, MO) and 1 × TrypLE Express Enzyme (Gibco, Grand Island, NY), and maintained in complete PEC medium (PECM) consisted of DMEM/F12 supplemented with 1 × insulin-selenium-transferrin (ITS; Gibco, Grand Island, NY), 5 ng/ml VEGF (PeproTech, Rocky Hill, NJ), 10 μM retinol, 4 μM RA (Stemgent, San Diego, CA) and 60 μg/ml ascorbic acid (Sigma). For plating, 1 μM of ROCK inhibitor Y-27632 dihydrochloride (Tocris Bioscience, Bristol, UK) was added for 24 h to facilitate cell attachment, after which medium was switched to PECM. For PEC maintenance, differentiated PEC were passaged and seeded at 10,000/well of 24-well plate and maintained using SB medium consisted of DMEM/F12 supplemented with 0.4 mg/ml of Albumax (Gibco, Grand Island, NY), 1 × ITS (Gibco, Grand Island, NY), 60 μg/ml ascorbic acid (Sigma Aldrich, St. Louis, MO), with 2 μM TGFβ inhibitor SB431542 (Tocris Bioscience, Bristol, UK). The proposed PEC differentiation was also repeated on two commercial lines, the Gibco hiPSC Episomal iPSC line reprogrammed from cord blood (Gibco, Grand Island, NY) and ATCC DYS0100 hiPSCs (ATCC ACS-1019) reprogrammed from foreskin fibroblast (ATCC, Manassas, VA). The same differentiation procedures and induction cocktails were used for both lines, but the CHIR99021 concentration was reduced to 6 μM for ATCC line during LPM induction.

**Stimulation of epithelial-mesenchymal transition in PECs.** Day 7 PECs were seeded at 100,000 cells/well on 24-well plate pre-coated with 0.1% gelatin, and allowed to attach overnight in basal medium (BM) consisted of DMEM/F12 supplemented with 0.4 mg of Albumax (Gibco, Grand Island, NY), 1 × ITS (Gibco, Grand Island, NY), 60 μg/ml ascorbic acid (Sigma Aldrich, St. Louis, MO). 1 μM of ROCK inhibitor Y27632 (Tocris Bioscience, Bristol, UK) was added to facilitate PEC attachment for 24 h. Then, PECs were expanded to full confluency in the BM with 2 μM SB431542 (Stemgent, San Diego, CA) but without AlbuMAX II (Gibco, Grand Island, NY), for 3-4 days. EMT were initiated by adding 5 ng/ml TGFβ (R&D system, Minneapolis, MN) for the initial 3 days and 10 ng/ml bFGF (PeproTech, Rocky Hill, NJ) for the subsequent 3 days or just bFGF alone, up to 6 days. Controls were epicardial cells in basal medium or with the addition of 2 μM SB431542 (Stemgent, San Diego, CA).

**Generation of mCherry or venus labeled iPS cells.** Lipofectamin 2000 (Invitrogen, Carlsbad, CA) was used for the transfection of 293 T cells with lentivirus vectors carrying either mCherry or Venus gene and packaging vectors. Medium of transfected 293T cells was collected 72 h after transfection, then filtered with 0.45 μm filter and concentrated. BJRiPS cells were trypsinized into single cells then infected with virus medium for 8 h. Infected cells were expanded for two rounds of FACS sorting to get a pure labeled population over 99%.

**Quantitative PCR.** Cells were lysed with RT buffer and total RNA was purified using RNeasy® Plus Mini kit (Qiagen, Hilden, Germany) which was reverse transcribed to cDNA using Superscript IV VILO Master Mix (Gibco, Grand Island, NY) and ran on Bio-rad T100$^{TM}$ Thermal Cycler according to 2manufacturer's protocol. Quantitative PCR was performed on StepOnePlus Real-time PCR system (Applied Biosystems, Foster City, CA) after mixing cDNA with Taqman Gene Expression Master Mix (Cat#: 4369016) and gene Taqman probe (Applied Biosystems, Foster City, CA) which can be found in Supplementary Table 1. All gene expressions were normalized to housekeeping gene beta-actin and presented as log$_2$fold change (delta-delta CT).

**Flow cytometry.** Cells were dissociated using trypsin, pelleted by centrifugation at 300 × g for 5 min, and fixed with Fixation/Permeabilization Solution Kit according to manufacturer's protocol (BD Biosciences, San Diego, CA). Cells were washed with two times of 10% BD Perm/Wash Buffer (BD Biosciences, San Diego, CA) to remove fixative prior to staining. Samples were incubated with primary antibody diluted in BD Perm/Wash Buffer (see Supplementary Table 2 for dilutions) at 4 ºC

for 45 min. To remove excessive primary antibody, samples were spun at $250 \times g$ for 5 min and washed once with Perm/Wash buffer prior to labeling with Alexa Fluor 488 secondary antibody (Molecular Probes Eugene, OR) at 1:500 for 30 min. All samples were washed once with BD Perm/Wash Buffer prior to analysis with BD Accuri, BD FACS LSR Flow Cytometer (BD Biosciences, San Diego, CA) or NovoCyte Flow Cytometer (ACEA Biosciences, San Diego, CA). All flow cytometric analyses were performed using FlowJo 10.5.0 software.

**Immunofluorescence labeling and microscopy (cells, spheres, aggregates)**. Cultured cells were fixed with cold methanol at 4 ºC for 15 min, and then washed with PBS. CM-spheres, PEC-Spheres, and cardiac aggregates were fixed overnight with 4% paraformaldehyde, embedded in Histogel™ (Thermo Scientific, Kalamazoo, MI), dehydrated, and embedded in paraffin. Sections were cut at 5 μm, deparaffinized, and treated with a heat-mediated antigen retrieval technique that included a 20-min boilin 0.01 M citrate buffer (pH 7.0). Both cells and histological sections were labeled with primary antibodies (Supplementary Table 2) overnight at 4 ºC. Donkey anti rabbit AlexaFluor 488 and donkey anti-mouse AlexaFluor 546 and 647 (Molecular Probes Eugene, OR) were used as secondary antibodies at 1:500 and nuclei were counterstained with DAPI (Sigma Aldrich, St. Louis, MO). All labeled cells were visualized using Nikon Eclipse Ti fluorescence microscope (Nikon Instruments, Melville, NY), and processed with NIS element imaging software or ImageJ.

**CM sarcomere length measurement**. Control CM and PEC/CM were fixed and stained with sarcomeric α-activin after 6 days in 2D co-culture. Sarcomere length was measured manually using ImageJ based on images taken at ×20 magnification using Nikon Eclipse Ti fluorescence microscope (Nikon Instruments, Melville, NY).

**PEC-CM migration and co-culture**. In order to distinguish between CMs and PECs in co-culture, all CMs were differentiated from Venus-tagged BJR-iPS cells and maintained in RPMI supplemented with insulin-containing B27. CMs from day 15 to 30 cultures were dissociated with collagenase, detached from culture plate with TypLE express with 63U of DNase I ((Invitrogen, Carlsbad, CA) were purified from non-myocyte cells using Percoll gradient separation method as published prior to use[65]. Day 7 PECs were produced using mCherry-tagged BJR-iPS cells. For migration assay, 50,000 of purified CM or 100,000 differentiated PECs were seeded into two-well cell migration inserts (Ibidi USA, Fitchburg, WI) attached on 24-well plates. Inserts were removed after 24 h to allow complete cell attachment, and images were taken at baseline, 17 h and after 30 h. For CM-PEC co-culture, CMs were generated from Venus-tagged BJR iPS cells and were seeded at 250,000 cells per well in 24-well plate 2 days prior to seeding with mCherry-tagged PECs at 250,000 cells per well. The culture was maintained in insulin-supplemented B27 in RPMI and PECM at 1:1 ratio. Changes in CM morphology were monitored and captured after 8, 16, and 21 days.

PECs and CM were seeded onto the migration inserts (Ibidi USA, Fitchburg, WI) and allowed to attach for 24 h. Mitomycin C (10 μg/ml) please check was added for 2 h prior to lifting the migration inserts. Cell movement toward the generated gap were imaged and captured at designated time using Nikon Eclipse Ti Fluorescence Microscope. Area recolonized by PECs were analyzed using ImageJ.

**RNA sequencing**. Samples were preserved in Trizol Reagent (Invitrogen Life Technologies, Carlsbad, CA) and sent for RNA extraction and analysis by Girihlet Inc. RNA Integrity Number (RIN) was determined by RNA Nano Bioanalyzer. RNA library was generated using TruSeq RNA Library Prep Kit with 80 bp single-end illumina sequencing (illumina, San Diego, CA). All fastq files and processed data were uploaded to Gene Expression Omnibus (GEO) database (GSE148543)[66,67].

The sequencing data were uploaded to the Galaxy web platform and were preprocessed at the public server at usegalaxy.org[68]. Fastq files were read and trimmed by using Trimmomatic[69]. First 10 bases were clipped and sequences below the average Phred score of 30 within a sliding window of four bases were trimmed. Results from individual samples were aggregated using MultiQC[70]. Based on the analysis, samples EpiP1_1 (GSM4473367) consisted of sequences with exceptionally high GC content (70%) and skewness and short length (43–44 bp) when compared to the rest of the libraries (50–54% GC and 51–60 bp in length), thus were removed from further analysis. Sequences from passed samples were mapped to hg38 human reference genome using Burrows-Wheeler Alignment tool for short sequences (<100 bp)[71]. Generated BAM files were merged accordingly[72] and subsequently used to produce gene counts using featureCounts v1.6.4 based on simple Illumina analysis mode[73].

We used iDEP.90, an integrated web application for differential gene expression and functional ontology analyses[74]. Only genes with a minimum of 5 counts per million (CPM) in all libraries were considered[75]. A total of 2000 most variable genes were included for hierarchical clustering based on the correlation (average linkage) method. A cut-off Z score of 4 was used. DESeq2 (idep ver 0.92) was used to determine the differential gene expression between two groups with FDR cut-off of 0.05 and minimum fold-change of $2^{76}$. Enrichment analysis of differentially

expressed genes (DEGs) based on GO terms for biological processes was also performed[77].

To compare the transcriptomes of PECs and Epi[P1] to previously published human primary-derived epicardial cells, we downloaded the fastq files from GSE84085 deposited in GEO[32]. To avoid batch effects between the studies, the sequences were pre-processed and analyzed according to the same pipeline described above. The list of DEGs from this study (PEC vs Epi[H9] and Epi[P1] vs Epi[H9]) and Bao et al.[32]. (Epi[DONOR] vs Epi[SC]) were used for both hierarchical clustering, PCA, and enrichment analysis based in GO terms for biological processes.

**Differentiation of ventricular-like CM population**. Human iPS cells were maintained and differentiated to LPM with Stemolecule™ CHIR99021 for 48 h. BMS-189453 (1–2 μM, Sigma Aldrich, St. Louis, MO) was supplemented together with Stemolecule™ Wnt inhibitor IWP-4 (Stemgent, San Diego, CA) in RPMI-INS for 48 h starting day 3. Then, BMS-189453 treatment was continued after an additional 48 h after withdrawal of Stemolecule™ Wnt inhibitor IWP-4 in RPMI-INS until day 9. Culture medium was then switched to RPMI supplemented with insulin-containing B27 to further maintain the differentiated ventricular myocytes until use.

**Ventricular-CMs proliferation in PEC, HCF, and HUVEC co-culture**. Generated ventricular-CMs at day 25–30 were used in this experiment. Differentiated ventricular-CMs and TrypLE express solution containing 63U/ml DNase1, enriched by Percoll® PLUS (GE-Amersham Biosciences, Uppsala, Sweden) density gradient separation method[1] and reseeded at $2.5 \times 10^5$ cells/well onto 24-well plate in RPMI-INS + 10% FBS. Day 7 PECs were seeded at $2.5 \times 10^5$ cells/well after 2–4 h of VM seeding. Then, media were refreshed with 10% Matrigel supplemented with RPMI-INS: PECM without RA at 1:1 ratio. Medium was refreshed every two days up to 6 days. Proliferative ventricular-CMs were labeled fixed and permeabilized with Fixation/Permeabilization Solution Kit (BD San Diego, CA), labeled for mouse anti human cTnT (1C11) (Abcam, Cambridge, MA) and counter-labeled with Click-iT™ Plus EdU Alexa Fluor™ 647 Flow Cytometry Assay Kit (Invitrogen, Carlsbad, CA) according to manufacturer's protocol. EdU and cTnT double-positive cells were quantified with FACS LSRII flow cytometer (BD San Diego, CA). All analysis was performed using FlowJo software. *IGF2 ELISA*. Conditioned media from 6 days PEC-Ventricular CM co-culture was collected at day 6 and frozen at −40 ºC prior to analysis. IGF2 in conditioned medium was analyzed using Human IGF-II Quantikine ELISA kit (R&D systems, Minneapolis, MN) in accordance with manufacturer's protocol.

**IGF signaling in PEC-induced ventricular-CM Proliferation**. Ventricular-CMs were seeded at $2.5 \times 10^5$ cells/well onto 24-well plate in RPMI medium supplemented with B27 minus insulin + 10% FBS. VMs were allowed to attach and the media were refreshed after 24 h. *IGF1R Inhibition* assay: every day with RPMI-INS containing 0, 0.06, 0.125, 0.25, 0.5, and 1 μM Linsitinib for 3 days. Proliferative VMs were were labeled with cTnT antibody and Click-iT™ Plus EdU Alexa Fluor™ 647 Flow Cytometry Assay Kit (Invitrogen, Carlsbad, CA) prior to quantification using FACS LSRII flow cytometer (BD Bioscience, San Diego, CA) as mentioned previously.

**IGF2 expression in PECs in response to ventricular-CMs**. mCherry-PECs were seeded at $2.5 \times 10^5$ cells/well with increasing Venus-CM number (0, 63,000, 125,000, 250,000, 250,000, 500,000) in 24-well plate, with medium change every day, up to 6 days. mCherry-PECs were separated from Venus-CM using BD FACS Aria Cell Sorter (BD Bioscience, San Diego, CA). Sorted mCherry-PECs were collected for RNA isolation and analysis, as described in earlier section (Quantitative PCR).

**IGF2 ELISA**. Ventricular-CMs were seeded at $2.5 \times 10^5$ cells/well onto 24-well plate in RPMI medium supplemented with B27 minus insulin + 10% FBS. VMs were allowed to attach and the media were refreshed after 24 h. mCherry-PECs were seeded at $2.5 \times 10^5$ cells/well. Conditioned media were harvested from PECs, CM and PEC-CM co-culture after 48 h of incubation at 37 ºC and kept in −80 ºC until analysis. IGF2 protein were quantified using Human IGF-II/IGF2 Quantikine ELISA kit DG200(R&D Systems, Minneapolis, MN). Samples were diluted 2–4 folds and IGF-II were quantified based on a standard log/log curve-fit with mean absorbance reading on the $y$-axis against the concentration on the $x$-axis. The optical density of each samples was obtained using Synergy HTX multi-mode reader (BioTek, Winnooski, VT).

**Inhibition of RA signaling in PEC in transwell**. PECs were seeded at $2.5 \times 10^5$ cells/well in 24-well plate while $2.5 \times 10^5$ cells VM were cultured in the transwell insert. Both cells were cultured using PECM:RPMI-INS medium, with BMS-189493 (5 μM) only present in lower compartment with PECs, for 6 days. *IGF2* RNA was harvested from PECs for qPCR analysis.

**Mitochondria staining**. Mitochondria from isolated CMs, cultured either alone or in co-cultured with PECs, were stained using Mitochondrial Staining Kit (Abcam, ab176747). Dissociated CMs were resuspended in pre-warmed RPMI+B27 with 10%FBS stained with the dye working solution from the kit and incubate for 37 ºC for 30 min for 2 h. The stained CMs were washed DPBS twice and then analyzed using flow cytometer (BD LSRII).

**Measurement of sarcomere length**. Dissociated and sorted CMs were seeded onto Fibronectin-coated 8-well chamber slide, cultured for 5 days and then fixed with 4% PFA. Cells were stained with α-actinin (Creative Diagnostics, DCABH-9438) and Goat anti-Rabbit IgG (H+L) Highly Cross-Adsorbed Secondary Antibody, Alexa Fluor 488 (Invitrogen), and imaged were acquired using the A1R confocal microscopy (Nikon). Sarcomere length was measured by the distance between the intensity peaks using FIJI/ImageJ software (ImageJ 1.53c, NIH). For CMs from PEC/CM co-culture, 373 sarcomeres from 16 CMs were analyzed. For CMs from monoculture, 230 sarcomeres from 30 CMs were analyzed (three independent replicates).

**Cell coverage and 2D-culture aggregate height analysis**. Cell Coverage: To assess cell coverage, Venus-tagged CMs (vCMs) were seeded either alone (250,000 cells/well in a 24-well plate) or in co-culture with other cell types including PECs, HUVECs, and HCFs (250,000 vCMs, with 250,000 cells of the co-cultured cell-type). Coverage was measured by isolating the green, fluorescent vCM signal from each ROI, thresholding all images to the same degree to subtract background, creating binary images, and then measuring surface-area coverage of signal versus total ROI area. Images were recorded using a Nikon Eclipse TE200 microscope (Nikon, Melville, NY). Image post-processing and surface-area coverage measurements were done using ImageJ software. In analyses of all groups at day 8 and 16, cell coverage measurements were taken from broad-field areas (7.6 mm$^2$) in three independent experiments (Figs. 4ciii and 5c). Data presented as mean ± SD. 2D-culture Aggregate Height: For height analysis, cell coverage experiments were analyzed at day 8 for all groups (Fig. S3E) and at day 16 for CMs-alone vs. PEC-CMs (Fig. 4ciii). Venus-tagged CM (vCM) aggregates were initially identified using fluorescent microscopy, using a Nikon Eclipse TE200 microscope. Microscope setting were changed for light microscopy, and then z-plane analysis was performed to carefully dial into high-resolution views of the top and bottom of the 2D-culture aggregates. Z-plane distance between the microscope lens and sample was recorded at both top/bottom focal planes using Nikon NIS-Elements Advanced Research software (measured in μm), and aggregate heights were determined by taking the difference of these positions. $N = 12$ for CM-alone and PEC-CMs at days 8 and 16 (Fig. 4c), as 4 measurements in each of three independent experiments (data presented as mean ± SD). For expanded analysis of all groups at day 8 (Fig. S3E), $n = 15$ for all groups as 5 measurements in each of three independent experiments (data presented as mean ± SD).

For all groups at day 8 in F (across three independent experiments), $n = 15$ for both groups at day 16 (across three independent experiments). Data presented as mean ± SD.

**Contractility analysis**. Strain measurement. We compared CM-only culture (1,500,000 CMs/well in six-well plates) to PEC-CM co-culture (900,000 CMs/well with 1,250,000 PECs/well in six-well plates), HUVEC-CM co-culture 900,000 CMs/well with 625,000 HUVECs/well in six-well plates), and HCF (900,000 CMs/well with 625,000 HCF/well in six-well plates co-culture. Experimental groups were seeded onto collagen gels consisting of 2.0 mg/mL collagen (collagen type I, derived from rat tails), supplemented with 0.9 mg/mL Matrigel. At 14 days, we acquired high-speed videos of beating areas at 10x magnification using a Nikon Eclipse TE200 microscope, under pacing conditions (20 V, 0.7 Hz, 6.0 ms) using a C-Pace cell culture stimulator (IonOptix LLC, Milton, MA). Videos were acquired with a 1280 × 1024-pixel resolution at a frame rate of 20 fps for 20 s, and then deconstructed to image-stacks. Strain of contracting areas was evaluated on successive images using a high spatial resolution sub-pixel algorithm called high-density mapping (HDM)[38]. We have previously used the HDM method to assess contractile strain in cardiac applications in vitro and in vivo[41,78]. Multiple contractions were analyzed per area (≥3), and several areas were analyzed ($n = 7$ for each group), and averaged to determine area-strain ($dA/A$). Atomic force microscopy (AFM): After strain measurements, cell-seeded gel samples (CM only, CM-HUVEC, CM-PEC, and CM-HCF) were cut to 1 cm by 1 cm squares and mounted on a petri dish and then submerged in cell medium to allow for AFM sampling. An Asylum Research MFP-3D-BIO AFM was used to image cell-seeded gels and measure the elastic modulus. Sharp conical cantilever tips made of silicon nitride were used that had a nominal spring constant of 0.06 N/m (DNP; Bruker Nano Inc.) For each sample ($n = 6$ for each group), three force maps were taken at separate locations; each force map consisted of 16 individual force curves (although two of the maps in the CMPEC samples only had 8 force curves taken due to a lack of height in the sample). Each force curve was fit to a Hertzian model over a 400 nm indentation range to find the resulting Young's modulus value ($E$). Contractility: Using the formula $F = (E) (dA/A) (Area)$, we calculated contractility ($Force/Area$) of the CMs in each group based on our measurements for strain and modulus.

**Calcium transients**. We compared CM-only culture (1,500,000 CMs/well in six-well plates) to PEC-CM co-culture (900,000 CMs/well with 1,250,000 PECs/well in six-well plates), HUVEC-CM co-culture 900,000 CMs/well with 625,000 HUVECs/well in six-well plates), and HCF (900,000 CMs/well with 625,000 HCF/well in 6-well plates co-culture. At day 14, intracellular calcium [Ca2+]i transients were studied with ratiometric intracellular calcium indicator Fluo-3 acetoxymethyl ester (Fluo-3 AM) at 37 ºC. Cells were loaded with ~4.4 μmol/L Fluo-3 AM (Invitrogen, USA) in normal Tyrode's solution (NaCl 136 mM, KCl 5.4 mM, MgCl2 1 mM, CaCl2 1.8 mM, NaH2PO4 0.33 mM, HEPES 5 mM and dextrose 10 mM; pH 7.35 with NaOH) in the presence of 0.02% Pluronic F-127 for 10 min at room temperature, followed by a 30 minutes washout of Fluo-3 AM for its de-esterification. Fluo-3 fluorescence was obtained with a FITC filter set (excitation: HQ480, mirror: Q505LP, emission: HQ535/50 m; Chroma) and an X-Cite exact mercury arc lamp (Luman Dynamics) with a 50% output for illumination. Fluorescent images were recorded with a Nikon Eclipse Ti-U inverted microscope (Nikon Instruments Inc., Melville, in [Ca2+]i NY, USA), a NeuroCCDSM camera (RedShirtImaging), and the Neuroplex software (RedShirtImaging, Decatur, GA, USA). Calcium transients were analyzed with Neuroplex and Clampfit 9.2 (Molecular Devices Inc., Sunnyvale, CA, USA). Multiple transient curves were analyzed per area (3–4), and several areas were analyzed in each condition (≥15). Fluo-3 AM calcium transient tracings were presented as ΔF/F0, F0 is the baseline fluo-3 fluorescence at resting state. Cyclic calcium transients were analyzed to determine amplitude, transient upstroke velocity, and transient decay velocity with a monoexponential fit. Data presented as mean ± SEM.

**CM-sphere, PEC-sphere, and aggregate formation**. To generate spontaneously beating CM-spheres from hiPSCs in suspension culture[79]. Briefly, hiPSCs were cultured in spinner flasks at 45 rpm for 24 h to form spheres, and then cardiac differentiation was achieved by 24 h treatment with 12 μM CHIR followed by 48 h treatment with 12 μM IWP4, resulting in spontaneously beating CM-spheres by day 10. To generate PEC-spheres, freshly differentiated PECs at day 7 were dissociated into single cells and suspension-cultured in RPMI supplemented with insulin-containing B27 and PECM at 1:1 ratio in a spinner flask using CELLSPIN system (Integra Biosciences AG, Switzerland), and spun at 45 rpm at 37 ºC. PEC spheres were mostly visible after 24 h. To create spontaneously beating cardiac aggregates, CM-spheres and PEC-spheres were sampled (10–20 spheres per group) and cultured in one well of a 24-well ultra-low attachment polystyrene plate (Corning Incorporated, Kennebunk, ME). Wells were fed every 2 days with insulin supplemented B27 in RPMI and PECM at 1:1 ratio. Rotation culture was the ultra-low attachment polystyrene culture plates with CM spheres and PEC-spheres that were placed on GyroMini Nutating Mixer (Labnet) ran at a fixed rotation speed of 20 rpm under standard cell culture conditions. Structures self-assembled over 48 h, with PECs incorporating with CM-spheres. Calcium transient and histology were assessed after 10 days in culture.

**Measurement of sphere density, size, structure**. A total of PEC and CM spheres were measured from day 1 of its formation after culturing in spinner flask culture (herein indicated as day-7 in Fig. 6b), and a total PEC spheres of 1303 and total of CM sphere of 1316, were included over the course of 7 days to calculate structures/spheres prior to co-culturing with CM spheres, using FIJI/ImageJ software under 10x magnification ($n = 3$ for day −7 −5. −3 and −1). To calculate percent of spheres with structure and the number of structure/spheres, we examined a total of 10 PEC spheres at day −7, 13 at day −5, 23 at day −3 and 30 at day −1. For CM spheres, we examined a total of 30 spheres at each timepoint from day −7 to day −1. The average area of structure in PEC spheres were derived from 16 measured structures at day −7, 35 at day −5, 137 at day −3, and 82 at day −1. Whereas the average area of structure in CM spheres were derived from 70 measured structures at day −7, 115 at day −5, 125 at day −3, and 69 at day −1. All readings were measured using FIJI/ImageJ software (Image J 1.53c, NIH) based on images taken at each timepoint at 10x magnification under Nikon Eclipse TE200 light microscope.

On day 15, PEC spheres, CM spheres, and PEC/CM spheres were measured for sphere density (total nuclei per mm$^2$), average sphere area in mm$^2$ ($n = 3$, with total measured PEC spheres = 10, CM spheres = 10, PEC/CM spheres = 3), structure per sphere ($n = 3$, with total measured PEC spheres = 10, CM spheres ≥ 10 and PEC/CM spheres = 8), and structure size/spheres size ($n = 3$, with total measured structure of 25 for PEC spheres, $n = 4$ with total measured structure of 11 for CM spheres, and $n = 3$ with a total measured structure of 13 for PEC/CM spheres). In additions, the number of structures per PEC/CM sphere and the average area of structure was measured at day 5 ($n = 10$, with a total of 25 measured structures) and day 15 ($n = 3$, with a total of 13 measured structures). All data were presented in mean ± SEM.

**TUNEL stain**. TUNEL stain was performed according to the manufacturer's protocol (DeadEnd™ Fluorometric TUNEL System, Promega Corporation, Madison, WI)). Briefly, Spheres were collected and fixed in 4% paraformaldehyde, embedded in Histogel Specimen Processing Gel (Thermofisher Scientific, Waltham, MA) prior to embedding in paraffin. Spheres were sectioned at 5 μm each slice, deparaffinized using fresh xylene, rehydrated with 100, 95, 85, 70, 50% graded

ethanol and washed with 0.85% NaCl at room temperature prior to fixing with 4% formaldehyde solution. Then the sphere sections were washed with PBS and pre-treated with 20 μg/ml Proteinase K solution for 10 min, after which were washed with PBS prior to labeling with rTdT incubation better solution consisted of equilibrium buffer, nucleotide mix and rTdT enzyme as provided by the manu-facturer. All sphere sections were counterstained with DAPI to visualize nuclei and apoptotic cells with fluorescein-12-dUTP fluorescein (green) were identified using Nikon Eclipse TE200 fluorescence microscope.

**Hematoxylin and eosin stain.** Paraffin-embedded tissue sections were depar-affinized in oven at 56 C for 15 min and rehydrated with HistoClear solution (National Diagnostics, Atlanta, GA) twice for 5 min, twice with 100% Ethanol (Fisher Scientific, Fair Lawn, NJ) for 3 min, 95% ethanol for 3 min and wash with distilled water for 2 min prior to staining with Hematoxylin (Vector laboratories, Burlingame, CA). Then, the slides were rinsed with distilled water for 2 min and counter-stained with Eosin (Vector laboratories, Burlingame, CA) with 15 dips, after which were rinsed with 95% ethanol and distilled water. Dehydration of slides were performed with submerging the slides for 3 min in 100% ethanol, and twice with HistoClear for 3 min. Sections were mounted with Permount solution, cov-ered with coverslips for microscopic examination.

**Statistical analysis.** All box and whisker's plot presented 25th, 50th, and 75th percentiles minima and maxima. Dotplots for gene expression data and bar graphs are presented in mean ± SEM. All data were representation of three independent replicates (unless stated otherwise) and expressed in mean ± standard error of mean. Difference between groups was analyzed using unpaired two-tailed student $T$-test (for two groups) or one-way ANOVA with Dunnett (for comparing mean of a defined control) or Tukey post hoc test for multiple comparisons, and it was considered statistically significant when $p < 0.05$. Unpaired $T$-test with Welch's correction was used for groups with unequal variance, which was tested using Shapiro–Wilk test. All analyses were performed using GraphPad Prism 8.0.2.

**Reporting summary.** Further information on research design is available in the Nature Research Reporting Summary linked to this article.

## Data availability

The RNA sequencing data discussed in this publication have been deposited in NCBI's Gene Expression Omnibus[66] and are accessible through GEO Series accession number GSE148543. The raw data that support the findings of this study are available from the corresponding authors upon reasonable request.

## Code availability

The image acquisition and processing methods for high-density mapping can be found within this article and references, including the manuscript where the method was first described (Kelly et al.[38]). The algorithm is available upon request from the corresponding author.

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

## Acknowledgements

The present study was supported by a seed grant from the Harvard Stem Cell Institute and by the Charles and Sara Fabrikant MGH Research Scholarship. J.J.T. is a recipient of FRGS Grant (203.CIPPT. 6711640) from the Ministry of Education Malaysia. Kenji Miki is partly supported by grants from JSPS KAKENHI (Grant Number 18KK0461) and SENSHIN Medical Research Foundation. We thank the facility and technical support from HSCICRM Flow Cytometry Core Facility. We also thank Professor Sean Palecek for providing us the H9-derived epicardial cells for RNA sequencing study. We thank Will Linthicum of Worcester Polytechnic Institute (PhD candidate), for his assistance and expertise with atomic force microscopy. We thank the Ministry of Education Malaysia and Universiti Sains Malaysia for sponsoring J.J.T.'s research fellowship.

## Author contributions

Conceptualization, J.J.T., J.P.G., and H.C.O.; Methodology, J.J.T., J.P.G., X.L., T.W., K.J.H., and K.H.L.; Investigation, J.J.T., J.P.G., K.M., L.X., G.K., T.W., L.Z., and K.J.H.; Writing – Original Draft, J.J.T. and J.P.G.; Writing – Review & Editing, D.J.M. and H.C.O.; Funding Acquisition, H.C.O., J.J.T., and K.M.; Resources, H.C.O., and J.J.T.; Supervision, H.C.O.

## Competing interests

H.C.O. is founder and stockholder of IVIVA Medical, Inc. This relationship did not affect the content or conclusions contained in this manuscript. J.J.T. received research grants from Cryocord Sdn Bhd. and ALPS Global Holding, Malaysia. All funders have no role in any of the experiments presented in this manuscript. Others declare no competing interests.
