## [Peer Review File · Nature Communications]

Reviewers' comments:

Reviewer #1 (Remarks to the Author):

The authors have generated what they term as 'proepicardial' cells from human ESCs. Initially they generate a lateral plate mesoderm equivalent followed by patterning to a 'proepicardial' (PEC) population. The authors then characterise the PECs and show that they appear to enhance hPSC-derived cardiomyocyte maturation. Overall, the novelty of the manuscript is low, since there are already a number of different methods to make (pro-)epicardial cells from human pluripotent stem cells (hPSC). The interactions with cardiomyocytes is interesting but incomplete and needs further work to show the exact effects on cardiomyocytes. In addition, the quality of the data presented could be improved and some of the interpretation of the transcriptomic data and cell populations represented is not fully justified as discussed below.

Major comments:

1. Have the authors generated proepicardial or epicardial cells? What are the markers suggesting these are proepicardial rather than epicardial? Since much of the novelty in this manuscript derives from the interaction of these cells with cardiomyocytes, then surely the characterised cell are epicardial or epicardial-derived?
2. Fig 1C – authors claim most cells expressed ZO1 with <1% expressing SMA or CD31. Need flow cytometry or at least much better quality and higher mag images of ZO to support this.
3. Fig 2B – need higher mag image to show widespread ZO1 staining
4. Page 9 - Based on differentially expressed genes, authors suggest that "PECs are primitive epicardial progenitors, and able to form mature epicardial cells.". Are they suggesting the EpiP1 are 'mature epicardial cells'? There is no evidence (morphology, gene expression, functional data) to support this assertion. Indeed since EpiP1 is closer to H9 than PEC, then perhaps this is actually a de-differentiated cell population?
5. Figure 3Aii, the PCA analysis, is missing!
6. Online figure 4 – cardiomyocytes seem to all stain for SMA, so difficult to see any induction at edges.
7. Online fig 5 – authors claim most of migrating cells are SMA+ WT1+. However the images seem to show that most of the SMA+ cells are in fact WT1-
8. Migrating PECs were treated with mitomycin C – what is the effect of this treatment on the cells – cell death, change in PEC markers and properties?
9. Figure 5 – there appears to be an error on the flow data in B. Although the number of cells that are cTnT+/EdU+ increases with PEC coculture, the total cell number with PEC co-culture is much higher as reflected by the coloured densities in the bottom 2 quadrants. It seems unlikely therefore that the top right quadrant is 37.2% of total – can the authors provide the % for the other 3 quadrants? Indeed the % for CTnT+/Edu+ cells is only given as 16% in figure Ci – why the discrepancy?
10. Online fig 12 – effects of coculture of epicardial and cardiomyocyte 3D bodies on TUNEL staining needs to be quantified.

11. The characterisation of effects of PECs on cardiomyocytes are incomplete. Further analysis of the cardiomyocytes in terms of ultrastructure including mitochondria, sarcomere lengths, cell size and T-tubule formation, is required.

12. What is the utility of the spheroids described in the last section and what do these studies really tell us? These are not used to make any discovery or to test any hypothesis.

--

Reviewer #2 (Remarks to the Author):

GENERAL COMMENTS

Tan et al. in this manuscript first described the protocol to differentiate hiPSCs to proepicardial cells (PECs) by combining the previous described methods by Lian et al (PNAS 2012), Witty et al (Nat Biotech 2014) and Iyer et al (Development 2015) to first differentiate hPSCs to mesoderm and LPM progenitors by the GSK3 β inhibitor (CHIR) activating canonical Wnt signaling, followed by BMP4, RA and VEGF treatment for 4 days to differentiate the LPM to proepicardial cells (PECs) which are enriched WT1+ cells (86.8%) and expressing ZO1, but not cTnT, and few CD31 and SMA. Then the authors demonstrate that the PECs can undergo epithelial-to-mesenchymal transition induced by TGF β , bFGF or combined treatments to transform to SMA+ and fibroblast phenotypic cells. Co-culture experiments of the hiPSC-derived CMs and PECs in 2D cultures resulted in improved contractility, more mature appearing intracellular Ca transients, and more cell division. In 3D culture, the authors found that PECs and CMs formed electrically coupled and mechanically active structures with complex cellular composition and also luminal structures. The main strengths and novelty of this manuscript are the more rapid differentiation protocol for PECs and the demonstration of the multiple effects of hiPSC-PECs co-culture on hiPSC-CMs properties. However, there remain a number of important limitations to the work as currently presented:

1. The authors state in the Introduction, "Furthermore, engineered tissues created solely from CMs lack the cellular and structural complexity of native myocardium, which likely contributes to construct immaturity." This statement is misleading in that since the initial work of Eschenahagen and colleagues in 1990s, it has been recognized that some mesenchymal population is needed in engineered cardiac tissues in additions to CMs. Furthermore, at least one prominent recent study has shown that engineered cardiac tissue can achieve substantial maturity, see Ronaldson-Bouchard et al., 2018 Nature 556:239. To this reviewer, the issue is more related to the best source of non-cardiomyocyte population and secondly what interventions are additionally needed to promote maturation if any such as high frequency pacing used by Ronaldson-Bouchard.
2. The reproducibility and robustness of the differentiation protocol is not evident. Figure 1Aii shows flow cytometry suggesting 86.8% of the cells are WT1 positive at day 7, but this is from a single experiment. What was the average result and variability?
3. Fig. 1B, the normalization of the gene expression is not clear. In the figure legend it states the fold change was normalized to the level of hiPSCs, however, in OCT4, ISL1, NKX2-5, the expression of the LPM cells were all 1, suggesting normalization was not done to iPSCs rather LPM. The different y-axes scales for the first 3 genes relative to other genes needs to be better highlighted as it is easy to miss the X1000.
4. The authors do not evaluate 2 key genes in defining proepicardial cells and epicardial cells in their characterization of this cell populations – TCF21 and ALDH1A2 which have typically been used by earlier studies of in vitro differentiation and development studies. For example, ALDH1A2 has been found to be particularly useful in separating the proepicardial cells from epicardial cells by multiple prior studies, e.g. Witty et al., 2014 (your referenece 14).
5. Did the protocol work in more than one hPSC line and are the co-culture results similar across lines? This is critical to understand how robust the protocol is because it cannot be assumed what works in

one line will be directly transferable to other lines.

6. Why do the authors focus their co-culture and functional experiments on pro-epicardial cells (PECs) rather than epicardial cells? No rationale for this is provided, and this is an important issue because epicardial cells are typically the key cell type to interact with cardiomyocytes and play a role directly in heart development. The characterization of the proposed P1 epicardial cells is limited to the gene expression studies.

7. The RNA-seq gene expression is confusing on multiple levels in Fig. 3. Using "H9 cells" to represent the hESC line H9-derived EPCs is very misleading. H9 commonly used to represent the undifferentiated H9 ESCs, so this should be changed for clarity. Why are H9-EPCs the primary comparator anyway? There are human primary epicardial lines available which would seem a better comparator. Why does gene ontology analysis focus on differentially expressed genes between PECs and H9 EPCs? Shouldn't the gene ontology analysis focus on the primary studied cell population in the manuscript, the PECs and gene expression in those cells? Although differential gene expression analysis is provided, there is no description of specific up or downregulated genes. Furthermore, the availability of the RNA-seq data is not described.

8. Supplemental Figure 4 if images are shows a regression of Venus-labeled CMs as mCherry-labeled PECs advance. There is no clear 'invasion' by differentiating PECs vs. a reduction and loss of CMs. This seems contrary to predictions and the authors' description.

9. For Figure 4aiii, the manuscript states the mean thickness of CM-only cultures is 34.1 μm , but the Figure shows something closer to 80 μm . Which is correct? Also how was thickness measured? This is not in the methods or figure legend that I find. Furthermore, heights of 2-D cultures of CMs reaching 80 μm would be quite unusual compared to the published literature.

10. It is not clear the purity of the CM only population. Obviously, this is a critical variable as other cells with the CMs could be impacting the results. Are CMs purified in some way?

11. The different functional measurements seem to sort differently for the conditions tested. The force production and Ca transient are greatest in the CMs+PECs, but the time to peak transient is lowest (fastest) in all three co-culture conditions relative to CM only. Likewise decay time does not comparably sort. How do the authors reconcile these differences? What is the mechanisms of these differences in the 3D engineered tissue – more cardiomyocytes as there is evidence for proliferative effects in other studies, more maturation or both? Some mechanistic investigation of the observed functional effects is needed to understand how PECs are impacting the biology. Are these mechanisms akin to development and specifically how?

12. The combination of spheroids in data presented in Figure 6 are indeed more complex with cystic structures forming, but what are these? How are they related to epicardial biology? There is significant heterogeneity in these based on the authors presentation, so it is not clear what the utility is of this approach or what is learned.

SPECIFIC COMMENTS

1. The title is confusing – "...Development" generally only refers to processes occurring in an intact organism, not in vitro.

2. The LPM progenitors express KDR and PDGFRa (double positive cells). Fig. 1Ai showed KDR and PDGFRa single positive cells, which doesn't represent the double positive LPM progenitors. There will clearly be overlap, but unless double-labeled the degree of double-positive cells will not be defined.

3. Supplement Figure 3ii and iii panels should be the same size, magnification and shape to allow overlap of labeling to be readily evident.

4. Based on the manuscript text, Figure 4 C-D are on engineered 3D tissue and not co-culture as in the initial part of this figure, but that is not made clear in the legend for the figure.

--

Reviewer #3 (Remarks to the Author):

The manuscript entitled "Human iPS-derived Pro-Epicardial Cells Enhance Directed Cardiac Development" by Tan et al. described differentiation of iPSCs to cells expressing epicardial progenitor markers, and effect of these cells on iPSC-derived cardiomyocytes in 2D and 3D cultures. The co-culture approach and results are novel and address an important problem in the cardiac tissue engineering field – how to enhance proliferation and maturation of cardiomyocytes to generate more functional and representative in vitro cardiac models. The two dimensional co-culture study which shows enhancement of cardiomyocyte proliferation and calcium handling is the strength of the work. The differentiation process is clearly described and the paper is generally clear and well-organized. The differentiation protocol builds on prior work by Iyer et al. and Witty et al., but the robustness of this needs to be demonstrated. The 3D culture models are not well-integrated with the 2D story and the functional relevance of the potentially interesting observations regarding tissue structure are not clear. Overall this paper has some interesting new findings but would be strengthened by clarifying some key points.

Major Concerns:

1. The protocol to generate proepicardial cells appears to be based off work by Iyer et al. and Witty et al. The authors should more clearly describe similarities and differences to this prior work.
2. The field generally requires demonstration of differentiation processes in several cell lines to demonstrate robustness of the process. It's important to compare differentiation yield and purity as well as establish expression of appropriate markers and functionality in several additional cell lines.
3. The authors should clarify criteria for claiming a proepicardial state. What markers and/or functions define this state? How does it differ from epicardial cells?
4. Figure 1B shows an upregulation of cTnT (TNNT2) but the last sentence of page 5 indicates that no cTnT+ cells were detected in the differentiations. What is the explanation for this discrepancy?
5. Throughout the manuscript the number of experimental replicates (including how many replicates are using cells from different differentiations, as opposed to technical replicates) should be clearly indicated in the figure captions
6. The RNAseq results are potentially interesting but the analysis is underdeveloped. It would be useful to compare these to datasets from primary epicardial/proepicardial samples in addition to the hESC-derived epicardial cells. The results show that after one passage, the proepicardial cells seem to be more epicardial like, but there are still key differences. A deeper analysis of the differentially-expressed genes would be useful to better understand similarities and differences to hESC-derived epicardium and primary epicardium.
7. The scratch migration assay used in this study can be affected by cell proliferation. Was the increased healing caused by proliferation or migration?
8. The calcium handling and contractility results in the coculture are promising, but the conclusions of enhancing maturation require a more comprehensive analysis of cardiomyocyte phenotypes (e.g. cell morphology, sarcomere and myofibril organization, metabolism).
9. The proepicardial cells appear to be an unstable state since they significantly change gene expression patterns after only one passage. This complicates analysis of coculture results since it isn't clear whether the proepicardial cells or a derivative of these cell types affects the cardiomyocytes. It's important to assess what happens to the proepicardial cells in both the 2D and 3D culture, in addition to profiling the cardiomyocytes.
10. The conclusions regarding RA and IGF signaling are not well-supported by the data. The data are consistent with induction of CM proliferation by RA and IGF but there is no causative data presented. Important controls are missing or difficult to interpret. Linsitinib affects proliferation in the absence of PECs, so this is not PEC-specific. There needs to be evidence that IGF2 produced by PECs is what is stimulating CM maturation. More specific inhibition of IGF2 in the CMs and demonstration that exogenous IGF2 addition to CMs in the absence of PECs may help support the mechanistic conclusions.
11. The 3D spheroid data are not well-integrated into the 2D story. The conclusions would be strengthened by demonstrating PEC effects on proliferation and maturation in 3D as well as 2D. While

there appear to be morphological changes induced by the PECs in the cardiac spheres, their relevance is unclear. This section lacks concrete solutions related to tissue organization and function and is highly descriptive and speculative. Terms like "cellular complexity" and "luminal structures" are vague. 12. The title is not appropriate for the manuscript. There are no effects on cardiac development shown. The title should be reworded to more accurately represent the effects on cardiomyocyte phenotype.

Minor concerns:

1. The ability of epicardium to generate endothelial cells is debated. There are issues with the lineage tracing models used to reach conclusions of epicardial fates, and some animal models seem to have an epicardial contribution to the cardiac vasculature while others may not. The manuscript should be clearer about the state of this field.
2. The authors should use gene names when referring to genes and protein names when referring to protein. For example, Fig. 1B should be gene names.
3. In the co-culture results, the authors need to be clearer about what cell densities were used. For example, in monoculture controls was total cell density or individual cell type number maintained constant, compared to the cocultures?

Response to Reviewers' comments:

Reviewer #1 (Remarks to the Author):

The authors have generated what they term as 'proepicardial' cells from human ESCs. Initially they generate a lateral plate mesoderm equivalent followed by patterning to a 'proepicardial' (PEC) population. The authors then characterise the PECs and show that they appear to enhance hPSC-derived cardiomyocyte maturation. Overall, the novelty of the manuscript is low, since there are already a number of different methods to make (pro-)epicardial cells from human pluripotent stem cells (hPSC). The interactions with cardiomyocytes is interesting but incomplete and needs further work to show the exact effects on cardiomyocytes. In addition, the quality of the data presented could be improved and some of the interpretation of the transcriptomic data and cell populations represented is not fully justified as discussed below.

Major comments:

1. Have the authors generated proepicardial or epicardial cells? What are the markers suggesting these are proepicardial rather than epicardial? Since much of the novelty in this manuscript derives from the interaction of these cells with cardiomyocytes, then surely the characterised cells are epicardial or epicardial-derived?

We thank the reviewer for the comment. We confirm that we have generated proepicardial cells at day 7 of differentiation, as the cells expressed significantly lower epicardial markers *TBX18*, *ITGA4*, *UPK1B*, *GJA1* and *RALDH2*, and hence less mature than that of in extended culture (Figure S1I) and after cocultured with CM (Figure 4A), suggesting a more epicardial-like cells after day 7.

2. Fig 1C – authors claim most cells expressed ZO1 with <1% expressing SMA or CD31. Need flow cytometry or at least much better quality and higher mag images of ZO to support this.
3. Fig 2B – need higher mag image to show widespread ZO1 staining

For 2-3. We thank the reviewer for the suggestion. We have included an image of WT⁺ ZO1⁺ with higher resolution (Figure 1C) and added another image with higher magnification (Figure S1C) in the manuscript. We have also reworked our remarks pertaining to this evidence in the first sub-section of the Results. In addition, have also included a FACS analysis on CD31 in Figure S1D.

4. Page 9 - Based on differentially expressed genes, authors suggest that “PECs are primitive epicardial progenitors, and able to form mature epicardial cells.”. Are they suggesting the EpiP1 are ‘mature epicardial cells’? There is no evidence (morphology, gene expression, functional data) to support this assertion. Indeed since EpiP1 is closer to H9 than PEC, then perhaps this is actually a de-differentiated cell population?

We apologise for the confusion. The H9 indicated in the original text was referring the H9-derived epicardial cells, which was obtained from Palecek's laboratory. We have re-named the H9 to Epi^{H9} to avoid further confusion. Epi^{P1} are extended culture of PECs for 1 passage with TGFβ inhibitor SB431542-supplemented medium, of which the phenotype and genotype are presented in Figure 2A and Figure 2B/S1I, respectively. We have also performed thorough RNASeq analysis to compare the gene expression profile between PECs, Epi^{P1} and Epi^{H9} (Figure 3), and the differential expressed genes (DEGs) of PECs and Epi^{P1} relative to Epi^{H9}, and the DEGs of bona fide donor epicardial cells (Epi^{Donor}) relative to Epi^{H9} as described on Page 5 (Figure S2).

5. Figure 3Aii, the PCA analysis, is missing!

We thank the reviewer for the comment. We have re-analysed the RNASeq data and included the PCA analysis in Figure 3B in the manuscript.

6. Online figure 4 – cardiomyocytes seem to all stain for SMA, so difficult to see any induction at edges.

We thank the reviewer for the comment. We have included a representative image of SMA/CM at edges at higher magnification in Figure S3D.

7. Online fig 5 – authors claim most of migrating cells are SMA⁺ WT1⁺. However, the images seem to show that most of the SMA⁺ cells are in fact WT1⁻

We thank the reviewer for the comment. We agree that it is inappropriate to make that claim without the backing of quantitative evidence. Nonetheless, we consistently observed more PEC-derived SMA⁺ cells in the migrated area, some of which retained WT1 expression (Figure S3D). We have amended the sentence in the text to better describe the observation.

8. Migrating PECs were treated with mitomycin C – what is the effect of this treatment on the cells – cell death, change in PEC markers and properties?

We thank the reviewer for the comment. To offset the proliferation effect of PECs during migration assay, we decided to use mitomycin C as a cell cycle antagonist, which has been extensively used in most cell migration assay (Simpson et al., 2013). We did not notice cell death at the concentration we used, and we were able to observe migratory cells that enabled us to take measurement on gap closure. It is possible that there were some changes in PEC properties and markers following mitomycin C treatment, but such changes/effects should have been normalized across groups, and should not affect our conclusion that PECs were more migratory in the presence of CM.

9. Figure 5 – there appears to be an error on the flow data in B. Although the number of cells that are cTnT⁺/EdU⁺ increases with PEC coculture, the total cell number with PEC co-culture is much higher as reflected by the coloured densities in the bottom 2 quadrants. It seems unlikely therefore that the top right quadrant is 37.2% of total – can the authors provide the % for the other 3 quadrants? Indeed the % for cTnT⁺/Edu⁺ cells is only given as 16% in figure Ci – why the discrepancy?

We thank the reviewer for pointing this out. We apologize for the mistake and have corrected it in the manuscript. The percentage is calculated based on the top two quadrants only and is relative to all cTnT⁺ cells (Figure 6E). We have drawn a box to make the figure more self-explanatory. We have also included a description in the figure legend as follow: The percentage was calculated based on 100% cTnT⁺ cells.

10. Online fig 12 – effects of coculture of epicardial and cardiomyocyte 3D bodies on TUNEL staining needs to be quantified.

We thank the reviewer for the suggestion. We agreed and have included the quantification of TUNEL staining in Figure 7C.

11. The characterisation of effects of PECs on cardiomyocytes are incomplete. Further analysis of the

cardiomyocytes in terms of ultrastructure including mitochondria, sarcomere lengths, cell size and T-tubule formation, is required.

We thank the reviewer for suggestion. We agree that the elucidation of the effects of PECs on the ultrastructural changes in CM would be a meaningful addition to our manuscript. We have added the measurement of sarcomere length in CM with and without PEC coculture (Figure S4). However, we are unable to perform further experiments due to several limitations involved in retroactive processing of samples and lab personnel turnover (including the departure of key members, now back at home institutions with more limited resources/capabilities).

12. What is the utility of the spheroids described in the last section and what do these studies really tell us? These are not used to make any discovery or to test any hypothesis.

PECs are known for their ability to attach and form an epicardial layer on the heart during development. Hence, we wanted to test if we could recapitulate this event and examine if such organized 'layering' epicardial cells on the outer surface of CM in three-dimensional culture could yield any advantage over making a construct with pre-mixed PEC/CM formula. In this process, we demonstrated that spherical PECs could attach to cardiac microtissue and promote synchronous contractility and formed a SMA⁺ layer on the outer surface of the cardiac microtissue. We also found the formation of luminal phenotype in PEC/CM microtissue indicative of vessel-like structure with defined SMA⁺ cell encoachment of VECad⁺ cells, of which is in line with the recent evidence presented by the Murry and Sinha laboratories (Bargehr *et al.*, 2019)(Bargehr *et al.*, 2019).

Reviewer #2 (Remarks to the Author):

GENERAL COMMENTS

Tan *et al.* in this manuscript first described the protocol to differentiate hiPSCs to proepicardial cells (PECs) by combining the previous described methods by Lian *et al.* (PNAS 2012), Witty *et al.* (Nat Biotech 2014) and Iyer *et al.* (Development 2015) to first differentiate hPSCs to mesoderm and LPM progenitors by the GSK3 β inhibitor (CHIR) activating canonical Wnt 3ignalling, followed by BMP4, RA and VEGF treatment for 4 days to differentiate the LPM to proepicardial cells (PECs) which are enriched WT1⁺ cells (86.8%) and expressing ZO1, but not cTnT, and few CD31 and SMA. Then the authors demonstrate that the PECs can undergo epithelial-to-mesenchymal transition induced by TGF β , bFGF or combined treatments to transform to SMA⁺ and fibroblast phenotypic cells. Co-culture experiments of the hiPSC-derived CMs and PECs in 2D cultures resulted in improved contractility, more mature appearing intracellular Ca transients, and more cell division. In 3D culture, the authors found that PECs and CMs formed electrically coupled and mechanically active structures with complex cellular composition and also luminal structures. The main strengths and novelty of this manuscript are the more rapid differentiation protocol for PECs and the demonstration of the multiple effects of hiPSC-PECs co-culture on hiPSC-CMs properties. However, there remain several important limitations to the work as currently presented:

1. The authors state in the Introduction, "Furthermore, engineered tissues created solely from CMs lack the cellular and structural complexity of native myocardium, which likely contributes to construct immaturity." This statement is misleading in that since the initial work of Eschenahagen and colleagues in 1990s, it has been recognized that some mesenchymal population is needed in engineered cardiac tissues in additions to CMs. Furthermore, at least one prominent recent study has shown that engineered cardiac tissue can achieve

substantial maturity, see Ronaldson-Bouchard et al., 2018 Nature 556:239. To this reviewer, the issue is more related to the best source of non-cardiomyocyte population and secondly what interventions are additionally needed to promote maturation if any such as high frequency pacing used by Ronaldson-Bouchard.

We thank to the reviewer's comment. We have amended our paragraph in the introduction to the following as suggested.

“Recent work showing CM engraftment and function in non-human primate and porcine models of heart failure reinforces the impact of this breakthrough (Chong et al., 2014; Kawamura et al., 2013). Despite these achievements, hPSC-derived CMs and resulting cardiac constructs remain phenotypically immature, with underdeveloped organization and electromechanical function (Veerman et al., 2015; Yang et al., 2014). Furthermore, bioengineered heart tissues using defined cardiac cells still lack the cellular and structural complexity of native myocardium, which could be greatly improved by implementing techniques that recapitulate and integrate important cues from heart development (Fong et al., 2016; Kim et al., 2010; Stevens et al., 2009).”

2. The reproducibility and robustness of the differentiation protocol is not evident. Figure 1Aii shows flow cytometry suggesting 86.8% of the cells are WT1 positive at day 7, but this is from a single experiment. What was the average result and variability?

We apologize for the misunderstanding. The 86.8% is an average WT1⁺ cells from three independent differentiation and analysis. We have amended the figure, and added more description in the text (Page 4, line 8) and the figure legend (Page 17).

3. Fig. 1B, the normalization of the gene expression is not clear. In the figure legend it states the fold change was normalized to the level of hiPSCs, however, in OCT4, ISL1, NKX2-5, the expression of the LPM cells were all 1, suggesting normalization was not done to iPSCs rather LPM. The different y-axes scales for the first 3 genes relative to other genes needs to be better highlighted as it is easy to miss the X1000.

Thanks for pointing this out. We have amended the mistake and highlighted the scale in red to make it clearer.

4. The authors do not evaluate 2 key genes in defining proepicardial cells and epicardial cells in their characterization of this cell populations – TCF21 and ALDH1A2 which have typically been used by earlier studies of in vitro differentiation and development studies. For example, ALDH1A2 has been found to be particularly useful in separating the proepicardial cells from epicardial cells by multiple prior studies, e.g. Witty et al., 2014 (your reference 14).

We thank the reviewer for the comment. We confirm that we have generated proepicardial cells at day 7 of differentiation, as the cells expressed significantly lower epicardial markers *TBX18*, *ITGA4*, *UPK1B*, *GJA* and *ALDH1A2*, and hence less mature than that of in extended culture (Figure S1) or after cocultured with CM (Figure 4A).

5. Did the protocol work in more than one hPSC line and are the co-culture results similar across lines? This is critical to understand how robust the protocol is because it cannot be assumed what works in one line will be directly transferable to other lines.

We thank the reviewer for the comment. We have included two additional cell lines to demonstrate the reproducibility of the differentiation protocol (Figure S1F and S1G).

6. Why do the authors focus their co-culture and functional experiments on pro-epicardial cells (PECs) rather than epicardial cells? No rationale for this is provided, and this is an important issue because epicardial cells are typically the key cell type to interact with cardiomyocytes and play a role directly in heart development. The characterization of the proposed P1 epicardial cells is limited to the gene expression studies.

We thank the reviewer for the comment. The rationale of using PECs in our study was to investigate if the PEO-to-heart event during development could be recapitulated *in vitro*. We demonstrated that the premature PECs were able to further differentiate and formed mature epicardial cells in CM coculture. Unlike other studies, we showed that PECs were able to adapt the presence of CM, and further improved CM contractile functions, proliferation, migration over the surface of 3D cardiac microtissue, integrated and developed organized luminal structure.

EpiP1 are extended culture of PECs for 1 passage with TGF β inhibitor SB431542-supplemented medium, of which the phenotype and genotype are presented in Figure 2A and Figure 2B/S1I, respectively. We have also performed thorough RNASeq analysis to compare the gene expression profile between PECs, EpiP1 and EpiH9 (Figure 3), and the differential expressed genes (DEGs) of PECs and EpiP1 relative to EpiH9, and the DEGs of bona fide donor epicardial cells (EpiDonor) relative to EpiH9 as described on Page 5 (Figure S2).

7. The RNA-seq gene expression is confusing on multiple levels in Fig. 3. Using "H9 cells" to represent the hESC line H9-derived EPCs is very misleading. H9 commonly used to represent the undifferentiated H9 ESCs, so this should be changed for clarity. Why are H9-EPCs the primary comparator anyway? There are human primary epicardial lines available which would seem a better comparator. Why does gene ontology analysis focus on differentially expressed genes between PECs and H9 EPCs? Shouldn't the gene ontology analysis focus on the primary studied cell population in the manuscript, the PECs and gene expression in those cells? Although differential gene expression analysis is provided, there is no description of specific up or downregulated genes. Furthermore, the availability of the RNA-seq data is not described.

We thank the reviewer for pointing this out. We have included new analysis on RNASeq in Figure 3 and S2.

8. Supplemental Figure 4 if images are shows a regression of Venus-labeled CMs as mCherry-labeled PECs advance. There is no clear 'invasion' by differentiating PECs vs. a reduction and loss of CMs. This seems contrary to predictions and the authors' description.

We thank the reviewer for the comment. We agree and could not exclude the possibility of having some CM loss as the mCherry-labelled PECs advance, though identification and quantification of dead CMs during the process presents several technical challenges. However, we noticed a prominent presence of mCherry+ cells (Figure S3iii) and WT1+ in the CM region (Figure S3Cix-x, S3D) at day 10 as compared to day 0 and day 1, suggesting of PEC invasion to CM region. Moreover, we also observed an increase in Venus intensity in some CM regions with network-like structure (Figure S3Ci-iii) and the presence of sarcomeric alpha-actinin positive CMs surrounded by SMA+WT1+ cells at the PEC/CM border (Figure S3D-new data), indicating the presence of functional CM at day 10 after PEC invasion.

9. For Figure 4aiii, the manuscript states the mean thickness of CM-only cultures is 34.1 μm , but the Figure shows something closer to 80 μm . Which is correct? Also how was thickness measured? This is not in the methods or figure legend that I find. Furthermore, heights of 2-D cultures of CMs reaching 80 μm would be quite unusual compared to the published literature.

We apologize for the mistake. The previous Figure 4aiiii has now moved to Figure 4Ciii. The correct measurement for thickness/aggregate height is $79.78 \pm 12.79 \mu\text{m}$. We have also included a description on the method of height measurement in the methodology.

10. It is not clear the purity of the CM only population. Obviously, this is a critical variable as other cells with the CMs could be impacting the results. Are CMs purified in some way?

We thank the reviewer for pointing this out, and we apologize for previously omitting this detail. We employed Percoll gradient centrifugation method in separating CM from the other non-myocyte cells. We have also included the description in the methodology under section: PEC-CM migration and co-culture (Supplemental Page 2).

11. The different functional measurements seem to sort differently for the conditions tested. The force production and Ca transient are greatest in the CMs+PECs, but the time to peak transient is lowest (fastest) in all three co-culture conditions relative to CM only. Likewise decay time does not comparably sort. How do the authors reconcile these differences? What is the mechanisms of these differences in the 3D engineered tissue – more cardiomyocytes as there is evidence for proliferative effects in other studies, more maturation or both? Some mechanistic investigation of the observed functional effects is needed to understand how PECs are impacting the biology. Are these mechanisms akin to development and specifically how?

We thank the reviewer for pointing out the differences in our calcium transient data: amplitude, time to peak, and decay time among the different conditions. We agree with the concern of the time to peak and decay time not comparatively sorting across groups. To reconcile this, we found it to be more appropriate to analyze the velocities of calcium transient upstroke and decay, to further highlight the kinetics of the recorded calcium transients. As shown in the following figures, PEC-cocultured CMs have significantly increased maximal upstroke (middle panel) and decay (right panel) velocities compared to CMs-alone, HUVEC-CMs, or HCF-CMs. Taken together with significantly higher Ca transient amplitude (left panel), these results suggest that co-cultured CMs may have more efficient calcium handling, especially in consideration of the enhanced mechanical contractility. Increased calcium transient amplitudes observed in PEC-CM co-culture could suggest that PEC-induced CMs have increased Ca^{2+} release by sarcoplasmic reticulum – a mechanism that could support enhanced inotropy. The mechanisms behind this augmented excitation-contraction coupling begs further investigation. Since enhanced force generation and calcium dynamics are indicators of improved hiPSC-derived CMs maturation (Yang et al., 2019), it is possible that the cardiomyocytes are undergoing enhanced developmental maturation when cocultured with PECs. We further showed that CMs in PEC co-culture demonstrated longer, more mature sarcomeres ($2.2 \pm 0.05 \mu\text{m}$) compared to CMs-alone ($1.8 \pm 0.03 \mu\text{m}$, $p < 0.001$; Figure S4), suggesting that PEC-CM co-culture could facilitate increased contractility through either improved length-dependent CM activation or enhanced CM maturity. However, more work is needed to understand the possible effects of PEC-CM cell interactions on CM morphology, sarcomere maturation, mitochondrial structure/function, and transverse tubule formation.

12. The combination of spheroids in data presented in Figure 6 are indeed more complex with cystic structures forming, but what are these? How are they related to epicardial biology? There is significant heterogeneity in these based on the authors presentation, so it is not clear what the utility is of this approach or what is learned.

PECs are known for its ability to attach and form an epicardial layer on the heart during development. Hence, we wanted to test if we could recapitulate this event and examine if such organized ‘layering’ epicardial cells on the outer surface of CM in three-dimensional culture could yield any advantage over making a construct with pre-mixed PEC/CM formula. In this process, we demonstrated that spherical PECs could attach to cardiac microtissue and promote synchronous contractility and formed a SMA+ layer on the outer surface of the cardiac microtissue. We also found the formation of luminal phenotype in PEC/CM microtissue indicative of vessel-like structure with defined SMA+ encoatment of VECad+ cells, of which is similar to the recent observation presented by Bargehr *et al.* (Simpson et al., 2013). Nonetheless, we did not observe higher proliferative CMs in PEC/CM microtissue.

SPECIFIC COMMENTS

1. The title is confusing – “...Development” generally only refers to processes occurring in an intact organism, not in vitro.

We thank the reviewer for the comment and have changed the title to: Human iPS-derived Pro-Epicardial Cells Direct Cardiomyocyte Aggregation, Expansion and Organization *In Vitro*.

2. The LPM progenitors express KDR and PDGFRa (double positive cells). Fig. 1Ai showed KDR and PDGFRa single positive cells, which doesn’t represent the double positive LPM progenitors. There will clearly be overlap, but unless double-labeled the degree of double-positive cells will not be defined.

We thank the reviewer for the comment. We amended our sentence to claim that the day 3 LPM cells are mostly KDR positive and/or PDGFRa positive (Page 4, line 6) as we agree that they are likely to have some overlay of double positive cells, which would better describe our observation.

3. Supplement Figure 3ii and iii panels should be the same size, magnification and shape to allow overlap of labelling to be readily evident.

We thank the reviewer for the comment. We have enlarged the panels (renamed as Figure S3Cvii-a, viii-a). In addition, we have also included a new representative image in Figure S3D to better show the labelling.

4. Based on the manuscript text, Figure 4 C-D are on engineered 3D tissue and not co-culture as in the initial part of this figure, but that is not made clear in the legend for the figure.

We thank the reviewer for the comment and apologise for the confusion. The previous Figure 4C and D (are now renamed as Figure 5D-H) are readings from CM coculture with PECs, HUVEC and HCFs in 2D, and not on the engineered 3D tissue. We have also amended the title of the figure legend for Figure 5 to “Effects of PECs on 2D CM co-culture” to better describe the figure.

--

Reviewer #3 (Remarks to the Author):

The manuscript entitled “Human iPSC-derived Pro-Epicardial Cells Enhance Directed Cardiac Development” by Tan et al. described differentiation of iPSCs to cells expressing epicardial progenitor markers, and effect of these cells on iPSC-derived cardiomyocytes in 2D and 3D cultures. The co-culture approach and results are novel and address an important problem in the cardiac tissue engineering field – how to enhance proliferation and maturation of cardiomyocytes to generate more functional and representative in vitro cardiac models. The two-dimensional co-culture study which shows enhancement of cardiomyocyte proliferation and calcium handling is the strength of the work. The differentiation process is clearly described and the paper is generally clear and well-organized. The differentiation protocol builds on prior work by Iyer et al. and Witty et al., but the robustness of this needs to be demonstrated. The 3D culture models are not well-integrated with the 2D story and the functional relevance of the potentially interesting observations regarding tissue structure are not clear. Overall this paper has some interesting new findings but would be strengthened by clarifying some key points.

Major Concerns:

1. The protocol to generate proepicardial cells appears to be based off work by Iyer et al. and Witty et al. The authors should more clearly describe similarities and differences to this prior work.

We agree with the reviewer and have included a paragraph in the discussion describing our similarities and differences with other published studies (Page 10).

2. The field generally requires demonstration of differentiation processes in several cell lines to demonstrate robustness of the process. It’s important to compare differentiation yield and purity as well as establish expression of appropriate markers and functionality in several additional cell lines.

We thank the reviewer for the comment and have performed the differentiation using the proposed protocol on two additional human iPSC lines (Figure S1F-G; 3 cell lines total).

3. The authors should clarify criteria for claiming a proepicardial state. What markers and/or functions define this state? How does it differ from epicardial cells?

We thank the reviewer for the comment. We confirm that we have generated proepicardial cells at day 7 of differentiation, as the cells expressed significantly lower epicardial markers *TBX18*, *ITGA4*, *UPK1B*, *GJA*

and *ALDH1A2*, and hence less mature than that of in extended culture (Figure S1) or after cocultured with CM (Figure 4A).

4. Figure 1B shows an upregulation of cTnT (TNNT2) but the last sentence of page 5 indicates that no cTnT⁺ cells were detected in the differentiations. What is the explanation for this discrepancy?

Thanks for the sharp observation. We have repeated the experiment 3 times and consistently observed an increase in *CTNT/TNNT2* genes in PEC at day 7, and confirmed that the increase did not translate into protein expression as demonstrated (Figure 1B-C). Nonetheless, when compared to the level of *CTNT/TNNT2* expression in differentiated CM, the expression in PEC was 235 times lower (Figure S1E).

5. Throughout the manuscript the number of experimental replicates (including how many replicates are using cells from different differentiations, as opposed to technical replicates) should be clearly indicated in the figure captions

Thanks for the reviewer's reminder. All experiments were performed with three independent biological replicates, using cells from different batches of differentiation. We have included this information in all figure legends.

6. The RNAseq results are potentially interesting but the analysis is underdeveloped. It would be useful to compare these to datasets from primary epicardial/proepicardial samples in addition to the hESC-derived epicardial cells. The results show that after one passage, the proepicardial cells seem to be more epicardial like, but there are still key differences. A deeper analysis of the differentially-expressed genes would be useful to better understand similarities and differences to hESC-derived epicardium and primary epicardium.

We thank the reviewer for pointing this out. We have included more robust RNASeq analyses of our samples, now presented in Figure 3 and Figure S2. All RNASeq data have also been uploaded to GEO.

7. The scratch migration assay used in this study can be affected by cell proliferation. Was the increased healing caused by proliferation or migration?

We have used mitomycin C to offset the proliferative effect (Simpson et al., 2013). Hence the observation and the analysis were based on the migratory function of PECs.

8. The calcium handling and contractility results in the coculture are promising, but the conclusions of enhancing maturation require a more comprehensive analysis of cardiomyocyte phenotypes (e.g. cell morphology, sarcomere and myofibril organization, metabolism).

We thank the reviewer for the thoughtful suggestion, and we believe that the suggested experiment would be a great addition, as well as a platform for future exploration. In this new version of the manuscript, we include an analysis of the velocities of calcium transient upstroke and decay, to further highlight the kinetics of the recorded calcium transients (Figure 5F and 5G). PEC-cocultured CMs have significantly increased maximal upstroke (middle panel) and decay (right panel) velocities compared to CMs-alone, HUVEC-CMs, or HCF-CMs. Taken together with significantly higher Ca transient amplitude (Figure 5E), these results suggest that co-cultured CMs may have more efficient calcium handling, especially in consideration of the enhanced mechanical contractility. Increased calcium transient amplitudes observed in PEC-CM co-culture could suggest that PEC-induced CMs have increased Ca²⁺ release by sarcoplasmic reticulum – a mechanism

that could support enhanced inotropy. The mechanisms behind this augmented excitation-contraction coupling begs further investigation. Since enhanced force generation and calcium dynamics are indicators of improved hiPSC-derived CMs maturation (ref Yang et al, 2019, Stem cell reports), it is possible that the cardiomyocytes are undergoing enhanced developmental maturation when cocultured with PECs. We also added a new analysis showing that CMs in PEC co-culture demonstrated longer, more mature sarcomeres ($2.2 \pm 0.05 \mu\text{m}$) compared to CMs-alone ($1.8 \pm 0.03 \mu\text{m}$, $p < 0.001$; Figure S4), suggesting that PEC-CM co-culture could facilitate increased contractility through either improved length-dependent CM activation or enhanced CM maturity. More work is needed to understand the possible effects of PEC-CM cell interactions on CM morphology, sarcomere maturation, mitochondrial structure/function, and transverse tubule formation. However, we regret we are unable to revisit all the experiments due to several limitations, ongoing lab member turnover, and the departure of key members to home institutions with limited resources and capabilities.

9. The proepicardial cells appear to be an unstable state since they significantly change gene expression patterns after only one passage. This complicates analysis of coculture results since it isn't clear whether the proepicardial cells or a derivative of these cell types affects the cardiomyocytes. It's important to assess what happens to the proepicardial cells in both the 2D and 3D culture, in addition to profiling the cardiomyocytes.

We thank the reviewer for this comment. We have included the gene and protein expression profiles of PEC after CM coculture in 2D (Figure 4A-B), and protein expression of PEC in 3D-CM microtissue culture (Figure 7D, S7).

10. The conclusions regarding RA and IGF signaling are not well-supported by the data. The data are consistent with induction of CM proliferation by RA and IGF but there is no causative data presented. Important controls are missing or difficult to interpret. Linsitinib affects proliferation in the absence of PECs, so this is not PEC-specific. There needs to be evidence that IGF2 produced by PECs is what is stimulating CM maturation. More specific inhibition of IGF2 in the CMs and demonstration that exogenous IGF2 addition to CMs in the absence of PECs may help support the mechanistic conclusions.

We thank the reviewer for the comment. Linsitinib is an inhibitor of insulin receptor and insulin-like growth factor 1 receptor (IR/IGF1R), and its ligand includes IGF2. We demonstrated that inhibition of IR/IGF1R using linsitinib decreased proliferative cardiomyocytes (Figure 6F). In the absence of supplemental IGF1 or insulin in the medium which could induce IR/IGF1R-mediated cell proliferation, CMs were found more proliferative in PEC coculture. This observation was coupled with increased *IGF2* expression and IGF2 secretion in PECs (Figure 6G), and the *IGF2* expression in PECs increased with increasing CM number (Figure 6H). Previous studies have shown that IGF2 is a known mitogen to induce CM proliferation during heart development (Shen et al., 2015), and also able to induce ESC- (McDevitt et al., 2005) or iPSC-derived CM proliferation (Kodo et al., 2016; Titmarsh et al., 2016). Hence, our data support that PEC-derived cells in CM coculture secreted IGF2 and contributed to CM proliferation.

11. The 3D spheroid data are not well-integrated into the 2D story. The conclusions would be strengthened by demonstrating PEC effects on proliferation and maturation in 3D as well as 2D. While there appear to be morphological changes induced by the PECs in the cardiac spheres, their relevance is unclear. This section lacks concrete solutions related to tissue organization and function and is highly descriptive and speculative. Terms like “cellular complexity” and “luminal structures” are vague.

We thank the reviewer for the comment. During heart development, a layer of epicardial epithelium is formed from cells migrated from proepicardial organ. The sequential events from the establishment of mature epicardium on the outer surface of the heart, and its subsequent differentiation and invasion to myocardium, contribute partly, if not solely, the final cellular diversity, ECM composition, CM density and overall architecture of a mature heart. Instead of mixing CM with PECs similar to the conventional method of generating engineered heart tissue, we asked if by recapitulating these PEO-to-heart developmental events *in vitro* using 3D cardiac spheroid culture could yield a structurally more organized and functionally better cardiac tissue. Here, we successfully demonstrated that PECs formed SMA⁺ cell layer on the surface of CM microtissue (Figure S7A), integrated in PEC/CM aggregates with promising excitation-contraction coupling with coupled [Ca²⁺]_i transients and contractile strain (Figure S5D). More interestingly, PECs also produced organized luminal, vessel-like structures in PEC/CM microtissue (Figure 7D) which was not seen in CM only microtissue. This observation is similar to a recent study that showed integration of SM22α⁺ epicardial cells into the vessel wall in the chorionic vasculature of chick embryos (Bargehr et al., 2019). Knowing the role of epicardial cells in coronary vessel formation during development, we show that the event could be recapitulated in 3D cardiac microtissue culture in the presence of PECs.

12. The title is not appropriate for the manuscript. There are no effects on cardiac development shown. The title should be reworded to more accurately represent the effects on cardiomyocyte phenotype.

We thank the reviewer for the comment and have changed the title to: Human iPS-derived Pro-Epicardial Cells Direct Cardiomyocyte Aggregation, Expansion and Organization In Vitro.

Minor concerns:

1. The ability of epicardium to generate endothelial cells is debated. There are issues with the lineage tracing models used to reach conclusions of epicardial fates, and some animal models seem to have an epicardial contribution to the cardiac vasculature while others may not. The manuscript should be clearer about the state of this field.

We thank the reviewer for the comments. We have performed FACS analysis on PECs and found that majority of the cells do not express CD31 (Figure S1D). However, we did notice the presence of CD31⁺ cells in day 7 PEC culture and the cells could persist in extended culture. Hence, we agree that we do not have sufficient evidence to show that PECs can become CD31⁺, and the endothelial cells that present in the microtissue was the cell by-product during PEC differentiation. We have also included this argument in the discussion (Page 12, Line 3-5).

2. The authors should use gene names when referring to genes and protein names when referring to protein. For example, Fig. 1B should be gene names.

We thank for the reviewer for pointing this out. We have corrected the gene name in Figure 1B, and amended the mistake throughout the manuscript.

3. In the co-culture results, the authors need to be clearer about what cell densities were used. For example, in monoculture controls was total cell density or individual cell type number maintained constant, compared to the cocultures?

We thank the reviewer for the comment. We used a constant CM cell density for all groups, with and without coculture with other cells, which was 2.5×10^5 cells. All the details have been stated in the methodology.

References:

1. Simpson MJ, Treloar KK, Binder BJ, Haridas P, Manton KJ, Leavesley DI, et al. Quantifying the roles of cell motility and cell proliferation in a circular barrier assay. *J R Soc Interface*. 2013;10(82):20130007.
2. Bargehr J, Ong LP, Colzani M, Davaapil H, Hofsteen P, Bhandari S, et al. Epicardial cells derived from human embryonic stem cells augment cardiomyocyte-driven heart regeneration. *Nat Biotechnol*. 2019;37(8):895-906.
3. Shen H, Cavallero S, Estrada KD, Sandovici I, Kumar SR, Makita T, et al. Extracardiac control of embryonic cardiomyocyte proliferation and ventricular wall expansion. *Cardiovasc Res*. 2015;105(3):271-8.
4. McDevitt TC, Laflamme MA, Murry CE. Proliferation of cardiomyocytes derived from human embryonic stem cells is mediated via the IGF/PI 3-kinase/Akt signaling pathway. *J Mol Cell Cardiol*. 2005;39(6):865-73.
5. Titmarsh DM, Glass NR, Mills RJ, Hidalgo A, Wolvetang EJ, Porrello ER, et al. Induction of Human iPSC-Derived Cardiomyocyte Proliferation Revealed by Combinatorial Screening in High Density Microbioreactor Arrays. *Sci Rep*. 2016;6:24637.
6. Kodo K, Ong SG, Jahanbani F, Termglinchan V, Hirono K, InanlooRahatloo K, et al. iPSC-derived cardiomyocytes reveal abnormal TGF-beta signalling in left ventricular non-compaction cardiomyopathy. *Nat Cell Biol*. 2016;18(10):1031-42.

REVIEWER COMMENTS

Reviewer #1 (Remarks to the Author):

This is a resubmission of a manuscript where the authors claim to have generated proepicardium from hPSCs, and shown a positive effect on cardiomyocyte maturation and function. Although they have answered some of the questions I raised previously, major issues remain regarding the cell type they have generated and the precise structural consequences on cardiomyocytes. In addition, the purely in vitro nature of their studies limits the significance of these findings.

Major comments:

1. There are now a number of protocols to generate epicardial cells from hPSCs. A major novelty claimed here is the derivation of proepicardium (PE). However, I don't think the authors are able to prove the cells are proepicardial as claimed. Indeed, the problem is that there are no clear markers of PE that are downregulated as the tissue starts to form an epicardial layer on the developing heart. The authors base their claim to have produced PE on lower levels of epicardial gene expression than after extended culture or after co-culture with cardiomyocytes. However, to my knowledge there are no robust in vivo comparative data comparing pro-epicardial gene expression levels to epicardial expression. It is just as likely that what they have made is a poorly differentiated epicardium (ie a pre-epicardium) rather than a pro-epicardium.
2. Figure 2 shows EMT and fibroblast differentiation (with bFGF) but no effort is made to investigate whether these cells form SMCs, another known cell type that epicardial cells can give rise to.
3. In figure 3, they have now clarified that H9 is in fact epicardium derived from H9 (Palacek lab). However, what is the point of doing this analysis, and showing that the PECs are less like EpiH9 than a passaged form (EpiP1) if they don't show any functional consequences. Does PEC have more of an effect on cardiomyocytes than EpiP1? Are there mechanistic pathways based on the RNAseq that may explain differences in function?
4. The characterisation of the effects of PEC on cardiomyocytes is still incomplete. The authors now provide data on increases in sarcomeric length with co-culture (fig S4), however the images supplied (presumably some of their better images) do not clearly demarcate individual sarcomeres and it would be impossible to calculate sarcomeric length from these data. Consequently the data in the bar chart is difficult to believe. In addition, there are still no data on other ultrastructural changes as previously requested (mitochondria, sarcomere lengths, cell size and T-tubule formation). The authors state that due to lab personnel turnover, these studies are not now possible. However co-culture of the 2 cell types and ultrastructural analysis is not a complex procedure and should be possible if the authors wish to validate their claims regarding the effects of PECs on cardiomyocytes.
5. Since I last reviewed this manuscript, Bargeher et al have published a paper (Nature Biotech 2019) showing both the in vitro and in vivo potential of hPSC-derived epicardium on cardiomyocytes, including in cardiac regeneration. I appreciate that asking the authors to carry out in vivo studies would be excessive, but inevitably the purely in vitro studies presented here are of reduced significance given the in vivo data that have emerged in this field.

Minor comments

1. Panels D and E are inverted in figure S1 legend.

--

Reviewer #2 (Remarks to the Author):

The authors present an extensively revised manuscript describing their protocol to generate pro-epicardial cells (PECs) from iPSCs and the effects of PECs when co-cultured with cardiomyocytes. The authors have directly and effectively addressed the major concerns raised in my initial critique. I have only the following minor concerns:

1. p. 7, "Contractility (force/area) was determined to be significantly enhanced for CMs in PEC-CM co-culture (0.029 ± 0.006 mN/mm², $p < 0.0001$ vs. all groups)" The authors measure strain elsewhere in the manuscript and give methods for that, but how do they actually measure force?
2. p. 9, "... kept viable und rotation culture..." What is rotation culture?
3. p. 10, "Collectively, these data confirm the identity of PECs and can become mature epicardial cells." The authors have not unambiguously demonstrated mature epicardial cells. By what criteria are the authors characterizing immature vs. mature epicardial cells. Maturation of all terminal lineages differentiated from iPSCs is a major challenge for the field, and it would be surprising if these cells did not also exhibit a degree of developmental immaturity. However, unlike cardiomyocytes, the metrics to determine epicardial cell maturity are not well defined. Therefore, this conclusion is tenuous.
4. The authors describe luminal structures present in their 3D cultures (Figure 7), but it is unclear what these luminal structures represent. What is the analogous structure in the native heart or embryo, given the goal of this methodology to model development?
5. As noted by the authors' following text from page 11 in the Discussion, "The mechanisms behind this augmented excitation-contraction coupling begs further investigation with respect to expression and function of key calcium channel proteins such as sarco/endoplasmic reticulum calcium-ATPase 2 (SERCA2), L-type calcium channel (LTCC) and Na⁺/Ca²⁺ exchanger (NCX), ryanodine receptor 2 (RyR2), and phospholamban50. In addition, potential evidence for transverse tubule organization and junctophilin-2 expression would likely suggest CM maturation 51. Further, investigating the expression of atrial markers (e.g. MLC1a, sln, Gja5) and ventricular markers (e.g. MLC1v, phospholamban) will help to determine if CM selection plays a role in electromechanical outputs³⁸." – there are obvious future mechanistic questions to pursue, which could be expanded on in more detail in the discussion.

--

Reviewer #3 (Remarks to the Author):

In this revision the authors made some improvements to clarify concerns raised by the reviewers. Some issues remain unaddressed or unclear:

1. The authors suggest that proepicardial cells are distinguished from epicardial cells by lower expression of epicardial-associated genes (e.g. TBX18, ITGA4, UPK1G, GJA, and ALDH1A2). This is a step in the right direction as it shows that the cells are progressing to mature epicardial cells. However, the authors should explicitly define criteria that distinguish various states of differentiation in their process (lateral plate mesoderm, proepicardial, epicardial, epicardial-derived). Since the focus of this paper is on proepicardial cells, it's most important to clearly define these cells, and low expression of epicardial markers is not sufficient. What is "low" and how do the proepicardial cells differ from immature epicardial cells? Upregulation of ALDH1A2 is useful to document, but what is the threshold for a defining an epicardial cells. Furthermore, the analyses here rely mostly on gene expression which

is unable to identify heterogeneity in cell states in the population. These conclusions of cell state transitions should be validated by flow cytometry.

2. Do the PECs directly transition to stromal cells through an EMT or do they first enter an epicardial state? Better definition of cell states and tracking of these states would help the reader understand what cell types exist in the different experiments presented in this paper.

3. The legend in Figure S2 appears incorrect. Where are the EpiH9 samples?

4. There is clear line-to-line variability in WT1+ cell generation (Fig. S1F,G). The authors indicated that they modified the protocol in different lines. They should provide details on what modifications they used, and what differentiation parameters most affect the generation of WT1+ cells so that others may more easily adopt their protocol in their own lines of interest.

5. The most novel and impactful aspect of this study is the reported effects of the PECs on CMs (and vice-versa) during co-culture. The morphological effects of the PECs on the CMs are clearly described. The influence of PECs on CM contractility is a very interesting finding. The influence on calcium handling and sarcomere length is consistent with higher contractility. A more comprehensive analysis of CM maturation phenotypes (e.g. gene expression, morphology, electrophysiology, etc.) is important to perform.

6. The sarcomere staining in Fig S4 is not sufficient to delineate z-discs or their spacing.

7. The authors' data on IGF2 and its inhibition is consistent with IGF2 signaling mediating PEC-CM interactions, but these data fall short of conclusively demonstrating the effects of PECs on CMs goes through IGF2-IGFR1 pathway. IGF2 activates multiple receptors and lisitinib is not specific for IGFR1. The study is missing important controls for the effects of lisitinib treatment on proliferation of CMs in monoculture, the addition of exogenous IGF2 to monoculture (at the doses measured in co-culture), and the effects of IGF2/lisitinib on receptor and pathway activation.

8. The 3D sphere data remain largely descriptive, without a clear tie to the effects of the PEC-CM interactions on heart development. The authors suggest that the 3D spheres are more tissue-like and have improved functionality, but there is not sufficient data to support these conclusions. The VE-cadherin expressing cells are very interesting but they may arise from the contaminating cell population in the PEC differentiation rather than the PECs. The structures here are not very representative of heart tissue. There are large vessel-like lumens but no microvasculature evident. Furthermore, functional superiority of the PEC-CM spheres needs experimental support.

REVIEWER COMMENTS

Reviewer #1 (Remarks to the Author):

This is a resubmission of a manuscript where the authors claim to have generated proepicardium from hPSCs, and shown a positive effect on cardiomyocyte maturation and function. Although they have answered some of the questions I raised previously, major issues remain regarding the cell type they have generated and the precise structural consequences on cardiomyocytes. In addition, the purely *in vitro* nature of their studies limits the significance of these findings.

Major comments

1. There are now a number of protocols to generate epicardial cells from hPSCs. A major novelty claimed here is the derivation of proepicardium (PE). However, I don't think the authors are able to prove the cells are proepicardial as claimed. Indeed, the problem is that there are no clear markers of PE that are downregulated as the tissue starts to form an epicardial layer on the developing heart. The authors base their claim to have produced PE on lower levels of epicardial gene expression than after extended culture or after co-culture with cardiomyocytes. However, to my knowledge there are no robust *in vivo* comparative data comparing pro-epicardial gene expression levels to epicardial expression. It is just as likely that what they have made is a poorly differentiated epicardium (i.e. a pre-epicardium) rather than a pro-epicardium.

We thank the reviewer for the comment. The true mature epicardium is formed only when PECs are attached to the heart *in vivo* where the microenvironment permits the exchange of developmental signalling between, and the growth of, the two cell types. Hence, PEC attachment on the myocardial surface, migrate and form an epicardial layer around the heart, drive CM proliferation, compaction and contribute to the formation of coronary plexus are known developmental events attributed to the function of the mature epicardium. In fact, some, if not all, of these events were successfully reproduced in our *in vitro* study using PECs. We also provided the first evidence that these cells can become epicardial cells when they were in contact with human CMs naturally, the closest mimic to the true *in vivo* conditions.

To answer the concern of the reviewer on the identity of pro-epicardium, we have referred to the most recent report by Cui et al. (2019) which demonstrated the single-cell transcriptomic analysis of the cardiac cells from human embryos ranging from 5-25 weeks.¹ In their study, proepicardial cells consisted of about 20% of all cardiac cells in the early human fetal heart at 5 weeks. We compared our PECs vs Epi^{H9} DEGs to their gene set from human proepicardial cells (hProEPs) and found that 4 out of 18 specifically upregulated genes in hProEPs were significantly overlapped with our PECs, namely the *HEY1*, *CRABP2*, *HAPLN1* and *HMGA2*; whereas *C1S*, the complement component predominantly expressed in mature epicardial cells, was among the downregulated DEGs in PEC vs Epi^{H9}. Furthermore, Cui also found that proepicardial cells were the most actively cycling cells among the cardiac cells, which also coincide with our results in GO term analysis comparing the DEGs of PECs vs Epi^{H9}. Noteworthy, the comparator data was derived from single-cell sequencing, which could explain the profile difference vs. our data which were generated from bulk RNA sequencing from PEC which could potentially be a heterogeneous population, as previously reported by Katz, T *et al.*²

In view of limited evidence to identify bona fide proepicardium, we sought to recapitulate the developmental functions of proepicardium. Two of the most fundamental functions that proepicardial cells are capable of, but has not been demonstrated elsewhere using mature human epicardial cells, are the abilities to 1) form epicardial epithelium in the presence of CMs, and 2) secrete IGF2, a known epicardial-derived mitogen that contributes to the growth of developing heart. In our study, PECs successfully demonstrated both attributes while in co-culture with CMs. In addition, PECs appear to have an effect on early CM-tissue formation, demonstrating consolidation of CMs into aggregates in both co-culture and co-differentiation.

Nonetheless, we decided to rename the cells as proepicardial-like cells (PEC) to better describe the population in view of the lack of more definitive and robust proepicardial markers *in vivo*.

2. Figure 2 shows EMT and fibroblast differentiation (with bFGF) but no effort is made to investigate whether these cells form SMCs, another known cell type that epicardial cells can give rise to.

We appreciate the reviewer's comment, but the claim that "no effort is made to investigate whether these cells form SMCs" does not accurately reflect the evidence provided in our study. We have demonstrated and repeatedly confirmed that SMC (SMA⁺ CNN1⁺) is one of the PEC fates, especially in the absence of TGFβ inhibition (Figure S1H), in 2D (Figure S3D) and 3D CM coculture (Figure 7). We have also reported SMA⁺ cells from PECs upon its sphere formation (Figure S6B), and the first to report the capability of PECs to form a layer of SMC⁺ cells over the surface of CM microtissue in 3D co-culture (Figure 7). We have also included images showing SMA⁺ TCF21⁺ cells in re-isolated PECs from CM coculture (Figure S1I).

3. In figure 3, they have now clarified that H9 is in fact epicardium derived from H9 (Palacek lab). However, what is the point of doing this analysis, and showing that the PECs are less like EpiH9 than a passaged form (EpiP1) if they don't show any functional consequences. Does PEC have more of an effect on cardiomyocytes than EpiP1? Are there mechanistic pathways based on the RNAseq that may explain differences in function?

Our focus in the manuscript has always been on the differentiation of PECs, and understanding the effects of PEC interactions with CMs. Expanding PECs beyond day 7 was merely an attempt to further characterize their identity, by comparing similarities and differences with the established epicardial cells. In this instance, the comparison was analysed using the Epi^{H9}, the H9-derived epicardial cells provided by Palecek's lab. We have also replicated the protocol by Bao *et al.* using PECs and showed that SB or A8301, the TGFβ inhibitors, could further expand day 7 PECs in serum-free medium up to passage 5, with favorable WT1⁺ expression. ³

4. The characterisation of the effects of PEC on cardiomyocytes is still incomplete. The authors now provide data on increases in sarcomeric length with co-culture (Fig S4), however the images supplied (presumably some of their better images) do not clearly demarcate individual sarcomeres and it would be impossible to calculate sarcomeric length from these data. Consequently, the data in the bar chart is difficult to believe. In addition, there are still no data on other ultrastructural changes as previously requested (mitochondria, sarcomere lengths, cell size and T-tubule formation). The authors state that due to lab personnel turnover, these studies are not now possible.

However, co-culture of the 2 cell types and ultrastructural analysis is not a complex procedure and should be possible if the authors wish to validate their claims regarding the effects of PECs on cardiomyocytes.

We thank the reviewer for the comment. We have improved the quality of the CM sarcomere length analysis using confocal microscopy and increased sampling for a revised Figure 6A, showing a significant difference between CM groups. We have also analyzed the mitochondrial content of the CM groups, showing that CMs from PEC/CM co-culture have a higher mitochondrial density vs. CMs from monoculture. While we analyzed cell size, we did not observe a significant difference between co-cultured and monoculture CMs (Figure 5L). Changes in gene expression were also not significantly different (data not shown); this could be a limitation of the short coculture window (6 days) and the time needed to detect changes in CM development and maturation, or it could also be due to the effect of PEC-induced CM proliferation (Figure 6E) offsetting the signals of maturing CM in the co-culture.

5. Since I last reviewed this manuscript, Bargeher et al have published a paper (Nature Biotech 2019) showing both the *in vitro* and *in vivo* potential of hPSC-derived epicardium on cardiomyocytes, including in cardiac regeneration. I appreciate that asking the authors to carry out *in vivo* studies would be excessive, but inevitably the purely *in vitro* studies presented here are of reduced significance given the *in vivo* data that have emerged in this field.

We thank the reviewer for the comment and appreciate that the addition of *in vivo* data would add significance to our study. Our first submission of this manuscript was dated back in early 2017-2018, to explore the derivation of proepicardial-like cells (PECs), to recapitulate the developmental events especially on the effects of myocardial proliferation and compaction; unlike other published works whose focus were on mature epicardial cells. Although our data are purely *in vitro*, we are first to introduce the derivation and the application of PECs from induced-pluripotent stem cells, with a much simpler and shorter differentiation method. We have also demonstrated the capabilities of PECs, which have not been shown using mature epicardial cells (e.g. CM proliferation, aggregation, and enhanced contractility and calcium handling in PEC co-culture). Moreover, our protocol can be easily integrated into CM differentiation protocol to allow co-differentiation of PEC/CM under a single culture. We also showed that the co-existence of both PEC and CM during *in vitro* differentiation drives PEC/CM spontaneous spatial organization. These capabilities generate critically important findings about proepicardium (possibly neglected transitory cells, and founder cells of epicardium), that beg further investigation for the potential in heart regeneration and tissue engineering, in addition to the works that are exploring the potential of mature epicardial cells.

To our knowledge, the only *in vivo* study available that used mature, embryonic stem cell-derived epicardial cells/CM was the study by Bargeher et al. (2019). However, the cell type and the scope of this manuscript were fundamentally different compared to Bargeher, as we used PECs versus mature epicardial cells. The *in vitro* data demonstrated in this study is necessary for providing a foundation that certainly warrants future assessment of the regenerative effects and application of PEC *in vivo*, as well as the difference in functional potential as compared to mature epicardial cells.

Minor comments

Panels D and E are inverted in figure S1 legend.

We thank the reviewer for pointing this out and we have amended the error accordingly.

Reviewer #2 (Remarks to the Author):

The authors present an extensively revised manuscript describing their protocol to generate pro-epicardial cells (PECs) from iPSCs and the effects of PECs when co-cultured with cardiomyocytes. The authors have directly and effectively addressed the major concerns raised in my initial critique. I have only the following minor concerns:

1. p. 7, “Contractility (force/area) was determined to be significantly enhanced for CMs in PEC-CM co-culture (0.029 ± 0.006 mN/mm², $p < 0.0001$ vs. all groups)” The authors measure strain elsewhere in the manuscript and give methods for that, but how do they actually measure force?

We thank the reviewer for the comment. The approach and techniques we use to arrive at Contractility (F/A) are fully described in the Materials and Methods section of the Supplement, under the sub-section *Contractility*. The methods employed did not necessitate a direct force measurement. Here, we provide a brief description. **First, area strain (dA/A) was determined** by analyzing high-speed videos of contracting CMs from all experimental groups, evaluating successive images using a high spatial resolution sub-pixel algorithm called high-density mapping (HDM). Results for area strain are shown in Figure 5C. **After strain measurements, Young’s modulus (E) of the cell-seeded gels were determined** using atomic force microscopy (AFM) to produce multiple force maps of each gel, which were then fitted to a Hertzian model using MATLAB. Results for gel modulus are shown in Figure 5D. **Contractility (F/A) was then calculated**, derived from the formula $F=(E)(dA/A)(Area) \rightarrow (F/Area)=(E)(dA/A)$, and using our measurements for strain (dA/A) and modulus (E). Results for contractility are shown in Figure 5E.

2. p. 9, “... kept viable under rotation culture...” What is rotation culture?

Rotation culture is a method to culture CM/PEC spheres in wells of a 24-well ultra-low attachment polystyrene plate under constant nutation, using a low-speed gyrating mixer set to a fixed rate of 20 rpm (GyroMini Nutating Mixer, Labnet). We have exchanged the word “rotation” with “nutation” for better clarity.

3. p. 10, “Collectively, these data confirm the identity of PECs and can become mature epicardial cells.” The authors have not unambiguously demonstrated mature epicardial cells. By what criteria are the authors characterizing immature vs. mature epicardial cells. Maturation of all terminal lineages differentiated from iPSCs is a major challenge for the field, and it would be surprising if these cells did not also exhibit a degree of developmental immaturity. However, unlike cardiomyocytes, the metrics to determine epicardial cell maturity are not well defined. Therefore, this conclusion is tenuous.

We agree with the reviewer's concern on the epicardial cell maturity and the lack of direct evidence to support the identity. In this study, we demonstrated:

1. The growth response and immunophenotype of PEC-derived EpiP1 expressing ZO1 and WT1 markers with low/no SMA differentiation in the presence of TGF β inhibitors A8301 or SB431542 (Figure 3A, S1H), similar to the previous reports.^{3,4}
2. The growth and extended culture of TGF β inhibitors-stimulated PEC showed further upregulation of epicardial markers TBX18, UPK1B, GJA and ITGA4. (Figure 1SI).
3. Extended PEC in PECM or CM co-culture showed *ALDH1A2* upregulation, the gene of the key regulatory enzyme for retinoic acid synthesis, which is known to be expressed in the epicardium (Figure4A, 1SI).

Nonetheless, we agree that these findings might not significantly demonstrate the maturity of the epicardium, but should suffice to show that PEC can be epicardial-like cells. Hence, we have revised our conclusion to indicate the differentiation towards more epicardial-like cells.

4. The authors describe luminal structures present in their 3D cultures (Figure 7), but it is unclear what these luminal structures represent. What is the analogous structure in the native heart or embryo, given the goal of this methodology to model development?

We thank the reviewer for the comment. The luminal structure in the 3D CM culture showed a rather different cellular orientation than what has been previously reported in the presence of CM. We found that the lumen showed an inner lining of VE-Cad⁺ cells with an outer SMA⁺ cell layer surrounding it which mimic a vessel-like structure in the presence of PECs. As the size of the lumen in the microtissue was too small thus posed technical challenges to further examine the function of the lumen and test if PECs involved in constructing vessel, as previously reported. As the evidence remains empirical, we de-emphasized the impact by moving the figure to the Supplement, in Figure S7.

5. As noted by the authors' following text from page 11 in the Discussion, "The mechanisms behind this augmented excitation-contraction coupling begs further investigation with respect to expression and function of key calcium channel proteins such as sarco/endoplasmic reticulum calcium-ATPase 2 (SERCA2), L-type calcium channel (LTCC) and Na⁺/Ca²⁺ exchanger (NCX), ryanodine receptor 2 (RYR2), and phospholamban50. In addition, potential evidence for transverse tubule organization and junctophilin-2 expression would likely suggest CM maturation 51. Further, investigating the expression of atrial markers (e.g. MLC1a, sln, Gja5) and ventricular markers (e.g. MLC1v, phospholamban) will help to determine if CM selection plays a role in electromechanical outputs³⁸." – there are obvious future mechanistic questions to pursue, which could be expanded on in more detail in the discussion.

We thank the reviewer for the comment and have added more information relevant to CM maturation in the discussion at the end of this paragraph in regards of electrical maturation: "The

effects of PEC-CM co-culture on CM electrical maturation are not clear. Future electrophysiological studies on isolated CMs from PEC-CM co-culture will be necessary to support the enhanced CM electromechanical function in the presence of PEC.”

Reviewer #3 (Remarks to the Author):

In this revision the authors made some improvements to clarify concerns raised by the reviewers. Some issues remain unaddressed or unclear:

1. The authors suggest that proepicardial cells are distinguished from epicardial cells by lower expression of epicardial-associated genes (e.g. *TBX18*, *ITGA4*, *UPK1G*, *GJA*, and *ALDH1A2*). This is a step in the right direction as it shows that the cells are progressing to mature epicardial cells. However, the authors should explicitly define criteria that distinguish various states of differentiation in their process (lateral plate mesoderm, proepicardial, epicardial, epicardial-derived). Since the focus of this paper is on proepicardial cells, it's most important to clearly define these cells, and low expression of epicardial markers is not sufficient. What is “low” and how do the proepicardial cells differ from immature epicardial cells? Upregulation of *ALDH1A2* is useful to document, but what is the threshold for defining an epicardial cells. Furthermore, the analyses here rely mostly on gene expression which is unable to identify heterogeneity in cell states in the population. These conclusions of cell state transitions should be validated by flow cytometry.

We thank the reviewer for the comments. We have included a schematic diagram to explicitly define the stage of differentiation to define lateral plate mesoderm and proepicardial cells (Figure 1A). We have also included a new figure to show the expression profiles for each cell stage represented in this work (Figure 7C).

To answer the concern of the reviewer on the identity of pro-epicardium, we have referred to the most recent report by Cui et al. (2019) which demonstrated the single-cell transcriptomic analysis of the cardiac cells from human embryos ranging from 5-25 weeks.¹ In their study, proepicardial cells consisted of about 20% of all cardiac cells in the early human foetal heart at 5 weeks. We compared our PEC vs Epi^{H9} DEGs to their gene set from human proepicardial cells (hProEPs) and found that 4 out of 18 specifically upregulated genes in hProEPs were significantly overlapped with our PECs, namely the *HEY1*, *CRABP2*, *HAPLN1* and *HMGA2*; whereas *C1S*, the complement component predominantly expressed in mature epicardial cells, was among the downregulated DEGs in PEC vs Epi^{H9}. Furthermore, Cui also found that proepicardial cells were the most actively cycling cells among the cardiac cells, which also coincide with our results in GO term analysis comparing the DEGs of PECs vs Epi^{H9}. Noteworthy, the comparator data was derived from single-cell sequencing, which could explain the discrepancies as our data which were generated from bulk RNA sequencing as PEC could have been a heterogeneous population as previously reported.

In view of the limited evidence to identify bona fide proepicardium, we sought to recapitulate the developmental functions of proepicardium. Two of the most fundamental function that proepicardial cells are capable of, but has not been demonstrated elsewhere using mature human epicardial cells, are the abilities to 1. form epicardial epithelium in the present of CMs, and 2.

Secrete IGF2, a known epicardial-derived mitogen that contributes to the growth of the developing heart. Both attributes were successfully demonstrated in this study when PECs were in CM coculture. Moreover, we also found that PECs were able to consolidate CM into aggregates.

Nonetheless, we decided to rename our differentiated cells as proepicardial-like cells (PEC) to better describe the population in view of the lack of more definitive and robust proepicardial markers *in vivo* to date.

2. Do the PECs directly transition to stromal cells through an EMT or do they first enter an epicardial state? Better definition of cell states and tracking of these states would help the reader understand what cell types exist in the different experiments presented in this paper.

In our hands, PECs were able to undergo EMT to derive mesenchymal/ stromal cells (Figure 2AB). This phenomenon has also been advocated in previous studies⁵. In 2D CM coculture, PECs gave rise to ECAD⁺ epicardial epithelial cells, CNN1⁺ cells, SMA⁺ cells (Fig 3C, 4B), most of which retained their epicardial marker expression WT1 and TBX18 (Fig S3H).

3. The legend in Figure S2 appears incorrect. Where are the EpiH9 samples?

We have amended the figure legend. The comparison of the gene set was among PECs, Epi^{P1} and Epi^{donor}.

4. There is clear line-to-line variability in WT1+ cell generation (Fig. S1F,G). The authors indicated that they modified the protocol in different lines. They should provide details on what modifications they used, and what differentiation parameters most affect the generation of WT1+ cells so that others may more easily adopt their protocol in their own lines of interest.

We have reproduced the protocol using the exact same cocktails and differentiation time frame. As for the human iPSCs we obtained from ATCC, we lower the CHIR99021 to 6 μ M from 12 μ M, as we encountered a significant level of cytotoxicity using the latter dose. We have now included the modification in the Methods (sub-section: *hiPSC culture, maintenance and differentiation*).

In BJRIpS and ATCC lines, there appears a higher concentration of WT1-high cells on Day 7. Whereas, the GIBCO line shows a more equal distribution of WT1-high and WT1-low expression levels on Day 7. It is possible that the GIBCO line is responding slower to treatment, as the histogram at Day 7 looks more similar to the expression at Day 5 for BJRIpS/ATCC.

Furthermore, cell lines were derived from different source cells (fibroblasts – ATCC, BJRIpS; hematopoietic, umbilical cord CD34 cells – GIBCO), using different reprogramming method (BJRIpS and ATCC were virally transduced, GIBCO were derived via episomal method). These are known variables that could affect the differentiation efficiency^{6,7}, and hence explain the need for adjusting the dose of differentiation mitogen used in this experiment. Additionally, new evidence also showed that the hiPSC response to CHIR99021 signaling is also somewhat dependent on the cell cycle state of hiPSCs⁸.

5. The most novel and impactful aspect of this study is the reported effects of the PECs on CMs (and vice-versa) during co-culture. The morphological effects of the PECs on the CMs are clearly described. The influence of PECs on CM contractility is a very interesting finding. The influence

on calcium handling and sarcomere length is consistent with higher contractility. A more comprehensive analysis of CM maturation phenotypes (e.g., gene expression, morphology, electrophysiology, etc.) is important to perform.

We thank the reviewer for the comment. We have re-analyzed CM sarcomere length after PEC coculture using confocal microscopy and have included it in Figure 6A. We have also analysed the mitochondrial content of the CM after PEC coculture (Figure 6B). However, we did not observe a significant increase in cell size and changes in gene expressions (data not shown). This is probably due to the short coculture window (6 days) that limits CM development and maturation. Furthermore, it could also possibly due to the effect of PEC-induced CM proliferation (Figure 6E) that offsets the number of maturing CM in the coculture.

6. The sarcomere staining in Fig S4 is not sufficient to delineate z-discs or their spacing.

We have re-performed this analysis with improved techniques, using confocal microscopic images and increased sampling, and observed a similar improvement in CMs isolated from PEC/CM coculture (Figure 6A).

7. The authors' data on IGF2 and its inhibition is consistent with IGF2 signaling mediating PEC-CM interactions, but these data fall short of conclusively demonstrating the effects of PECs on CMs goes through IGF2-IGFR1 pathway. IGF2 activates multiple receptors and linsitinib is not specific for IGFR1. The study is missing important controls for the effects of linsitinib treatment on proliferation of CMs in monoculture, the addition of exogenous IGF2 to monoculture (at the doses measured in co-culture), and the effects of IGF2/linsitinib on receptor and pathway activation.

We thank the reviewer for the comment. We have repeated the experiment and found the dose-dependent decrease in EdU+ cTnT+ cells in response to increasing Linsitinib (OSI-906) concentrations PEC/CM coculture (Figure 6F). We have also confirmed that the addition of exogenous IGF2 increased EdU+ cTnT+ cells (Figure 6G). We agree that linsitinib is not a specific inhibitor for IGFR1, but also reacted to insulin receptor, glycogen synthase kinase-3 β (GSK-3 β), Ca²⁺/calmodulin-dependent protein kinase kinase II (CAMKK2), insulin receptor-related kinases, and casein kinase 2 α 2 (CK2 α 2)⁹.

To exclude the growth effect possibly induced by the insulin, we have omitted insulin in the culture medium for CM proliferation experiment, hence the off-target inhibition related to insulin receptor should not affect our observation. Glycogen synthase kinase-3 β (GSK-3 β) is another interesting linsitinib off-target inhibition, which could lead to an accumulation of beta-catenin and thus activating the Wnt signaling pathway. Nonetheless, studies have shown that Wnt activation and beta-catenin is required for cardiomyocyte proliferation^{10, 11}, which was not in line with our observed inhibitory effect in CM. Other off-target inhibitions by linsitinib such as CAMKK2 and CK2 α 2 have not shown to involve in cardiomyocyte proliferation, thus the inhibitory effects should not affect our conclusion.

8. The 3D sphere data remain largely descriptive, without a clear tie to the effects of the PEC-CM interactions on heart development. The authors suggest that the 3D spheres are more tissue-like and have improved functionality, but there is not sufficient data to support these conclusions. The VE-cadherin expressing cells are very interesting but they may arise from the contaminating cell population in the PEC differentiation rather than the PECs. The structures here are not very representative of heart tissue. There are large vessel-like lumens but no microvasculature evident. Furthermore, functional superiority of the PEC-CM spheres needs experimental support.

The 3D microtissue experiment was to show the high level of structural complexity with an organised luminal structure mimicking the vessel in the presence of PECs. We do not imply that the VE-cadherin expressing cells were derived from PECs and agree that it could come from the contaminating cells from PEC differentiation (Figure). We showed the cellular organization of the lumen was an effect of PECs in constructing vessel, but given the lumen size, demonstrating the vessel function poses significant technical challenges. We agree that the data remains empirical and have de-emphasized its impact by moving it to the Supplement, to Figure S7.

References

1. Cui Y, *et al.* Single-Cell Transcriptome Analysis Maps the Developmental Track of the Human Heart. *Cell Rep* **26**, 1934-1950 e1935 (2019).
2. Katz TC, *et al.* Distinct compartments of the proepicardial organ give rise to coronary vascular endothelial cells. *Developmental cell* **22**, 639-650 (2012).
3. Bao X, *et al.* Long-term self-renewing human epicardial cells generated from pluripotent stem cells under defined xeno-free conditions. *Nat Biomed Eng* **1**, (2016).
4. Moerkamp AT, *et al.* Human fetal and adult epicardial-derived cells: a novel model to study their activation. *Stem Cell Research & Therapy* **7**, 174 (2016).
5. Olivey HE, Mundell NA, Austin AF, Barnett JV. Transforming growth factor-beta stimulates epithelial-mesenchymal transformation in the proepicardium. *Dev Dyn* **235**, 50-59 (2006).
6. Hu S, *et al.* Effects of cellular origin on differentiation of human induced pluripotent stem cell-derived endothelial cells. *JCI Insight* **1**, (2016).
7. Paniza T, *et al.* Pluripotent stem cells with low differentiation potential contain incompletely reprogrammed DNA replication. *J Cell Biol* **219**, (2020).
8. Laco F, *et al.* Unraveling the Inconsistencies of Cardiac Differentiation Efficiency Induced by the GSK3beta Inhibitor CHIR99021 in Human Pluripotent Stem Cells. *Stem Cell Reports* **10**, 1851-1866 (2018).

9. Anastassiadis T, *et al.* A highly selective dual insulin receptor (IR)/insulin-like growth factor 1 receptor (IGF-1R) inhibitor derived from an extracellular signal-regulated kinase (ERK) inhibitor. *J Biol Chem* **288**, 28068-28077 (2013).
10. Buikema JW, *et al.* Wnt/beta-catenin signaling directs the regional expansion of first and second heart field-derived ventricular cardiomyocytes. *Development* **140**, 4165-4176 (2013).
11. Singh AP, Umbarkar P, Guo Y, Force T, Gupte M, Lal H. Inhibition of GSK-3 to induce cardiomyocyte proliferation: a recipe for in situ cardiac regeneration. *Cardiovasc Res* **115**, 20-30 (2019).

REVIEWER COMMENTS

Reviewer #1 (Remarks to the Author):

The authors have provided further data and improved the quality of their manuscript. However there are still some major doubts as to whether they have generated bona-fide pro-epicardium that to me are still not fully resolved as discussed below. Also, they now add real novelty in showing a new co-culture method which could be extremely useful for translational applications, but these data are very preliminary in nature.

Specific comments.

1. The authors have made an effort to compare their PEC to human 'pro-epicardial cells' identified in early human embryos from scSeq data (Cui et al). However, the cells identified by Cui et al are not pro-epicardial, since the earliest heart examined was from a human embryo at week 5. However, in human hearts, the epicardium has completely covered the heart by Carnegie stage 15 (week 5) (Risebro et al Development 2015). Therefore Cui et al incorrectly call these cells 'pro-epicardial' and this group of cells expressing epicardial markers should be more correctly termed 'early fetal epicardium'.

2. Nevertheless, the authors here compare their PECs to the hProEPs identified by Cui et al by looking at genes upregulated or downregulated in PEC vs EpiH9 and looking at the representation factor in hProEP genes. But these studies lack suitable controls. For example why not do a similar analysis for genes upregulated in EpiH9 vs PEC and calculate their representation factor with hProEP genes.

3. The authors suggest that functionally PEC behaves like proepicardium with epithelial formation and IGF2 expression with CM and effects on CM. Again these studies lack suitable controls. For example, do EpiH9s have a similar effect? Also the effects of PEC on CM are similar to the effects of hESC epicardial cells on cardiomyocytes previously demonstrated by Bargeher et al.

4. Comments on new data: The co-culture experiment illustrated in figure 4E is very interesting. However the authors did not demonstrate that the WT1+ population obtained this way is equivalent to the PEC they produce in mono-culture. The cells are only investigated for WT1+ expression and the results shown are n=1. The structure shown on the pictures is very interesting but difficult to understand. Are all the cells organizing in one tube or a network of tubes? A lower magnification of the plate or more details in the text would be suitable. Do the 'PEC' cells generated at the same time as the cardiomyocytes have the same, less or more positive effects on the cardiomyocytes? This is an interesting system which deserves better characterisation. If the co-development is as efficient as the mono-culture it should be implemented as it saves time and might also present other advantages.

Risebro CA, Vieira JM, Klotz L, Riley PR. Characterisation of the human embryonic and foetal epicardium during heart development. Development. 2015;142(21):3630-3636.
doi:10.1242/dev.127621

Reviewer #2 (Remarks to the Author):

The authors have adequately addressed my critiques. The additional data further strengthen the paper.

Reviewer #3 (Remarks to the Author):

The authors significantly revised their manuscript and addressed the vast majority of my comments. The major remaining concern is the lack of precision in defining a PEC. This is critical for reproducibility and precision of this work. It would have been very helpful to have had a version of the manuscript with changes highlighted. Also, the figure numbers in the response document didn't match the figure numbers in the manuscript. Thus it took a lot longer to review this revision than it should have.

1. The revision of the terminology to "proepicardial-like cells" is an improvement. The use of the Cui dataset to compare the proepicardial cells to epicardial cells in vitro is a good idea. I suggest that in light of this the authors revise Figure 1B to identify the genes that distinguish proepicardium from epicardium, including both proepicardial and epicardial genes in this panel. I agree with the authors that these are not the easiest cell types to distinguish (or to map onto cell types in vitro) but all of the genes they list in Figure 1B are also in epicardial cells. A clearer definition of what sets of markers uniquely identify the PECs from other cell types is crucial for reader understanding of the population used in this paper, and for others to isolate and identify this population in their work. Along these lines, Figure 7C shows a putative model of differentiation but as structured it's misleading. As stated in the next point, it's not conclusive that the PECs directly generate SMCs or fibroblasts. Epicardial cells can also generate these cell types. Also, some of the markers listed as PEC are also in epicardial cells but not listed there. I suggest removing this panel or at least substantially revising it to more accurately represent the roles of PECs AND epicardial cells.
2. Figure 3A and B don't demonstrate that the PECs directly transit to stromal cells without going through an epicardial state. However this is a tangential point to the core of the manuscript and not necessary to address further.
3. The sarcomere structure in Fig 5J is much easier to see. As a minor point, it would help to have a scale bar in Fig 5A. It's not easy for the reader to convert the provided area to length.

Response to Reviewers' Comments

Reviewer #1

The authors have provided further data and improved the quality of their manuscript. However there are still some major doubts as to whether they have generated bona-fide pro-epicardium that to me are still not fully resolved as discussed below. Also, they now add real novelty in showing a new co-culture method which could be extremely useful for translational applications, but these data are very preliminary in nature.

Specific comments.

1. The authors have made an effort to compare their PEC to human 'pro-epicardial cells' identified in early human embryos from scSeq data (Cui et al). However, **the cells identified by Cui et al are not pro-epicardial,** since the earliest heart examined was from a human embryo at week 5. However, in human hearts, the epicardium has completely covered the heart by Carnegie stage 15 (week 5) (Risebro et al Development 2015). Therefore Cui et al incorrectly call these cells 'pro-epicardial' and this group of cells expressing epicardial markers should be more correctly termed 'early fetal epicardium'.

We thank the reviewer for the comment. To prove the identity of the cells we generated in this manuscript, we think that there is no other good comparator besides the proepicardial cells which are isolated directly from proepicardial organ in the developing human embryos. Judging from the unavailability and ethical concerns that may arise from acquiring this control sample, we admit that it is impossible for us to provide further and stronger evidence in this regard. In view of the lack of this control and evidence from past studies and literature to distinguish the stage of the cells prior to epicardial cell formation, we have resolved to rename the term to better describe the cells as pre-epicardial cells (PECs), as suggested in the earlier review. We carry this change throughout the manuscript and provide a highlight in the Discussion (p. 11).

2. Nevertheless, the authors here compare their PECs to the hProEPs identified by Cui et al by looking at genes upregulated or downregulated in PEC vs EpiH9 and looking at the representation factor in hProEP genes. But these studies lack suitable controls. For example why not do a similar analysis for genes upregulated in EpiH9 vs PEC and calculate their representation factor with hProEP genes.

*We thank the reviewer for the comment. We have included this new control as seen in **Figure S2C** to bolster the rigor of the analysis, as suggested.*

3. The authors suggest that functionally PEC behaves like proepicardium with epithelial formation and IGF2 expression with CM and effects on CM. Again these studies lack suitable controls. For example, do EpiH9s have a similar effect? Also the effects of PEC on CM are similar to the effects of hESC epicardial cells on cardiomyocytes previously demonstrated by Bargeher et al.

We thank the reviewer for the suggestion. We do not exclude the possibility that the effects we have seen in CM could also be coming from differentiated epicardial cells from PECs, as they can become more mature alongside the presence of CM in the coculture. However, this suggests that using PECs could also offer similar functional effects on CMs as demonstrated by Bargeher et al. in addition to using mature epicardial cells.

4. Comments on new data: The co-culture experiment illustrated in figure 4E is very interesting. However the authors did not demonstrate that the WT1+ population obtained this way is equivalent to the PEC they produce in mono-culture. The cells are only investigated for WT1+ expression and the results shown are n=1. The structure shown on the pictures is very interesting but difficult to understand. Are all the cells organizing in one tube or a network of tubes? A lower magnification of the plate or more details in the text would be suitable. Do the 'PEC' cells generated at the same time as the

cardiomyocytes have the same, less or more positive effects on the cardiomyocytes? This is an interesting system which deserves better characterisation. If the co-development is as efficient as the mono-culture it should be implemented as it saves time and might also present other advantages.

We thank the reviewer for the comment. We have repeated the co-differentiation 3 times to confirm the reproducibility of the protocol. We agree that the image may not sufficiently be self-explanatory. We have included an image with a lower magnification to show the phenomenon in a larger area (Figure S3) and a movie video that show the contracting CMs after 10 days of co-differentiation (Movie S1). We appreciate that further in-depth characterization of the co-development system could be helpful, but a full analysis and comparison is out of scope for this manuscript. We anticipate follow-up studies to give this more thorough consideration, especially to better characterize cellular interactions over time and explore potential advantages (or limitations) more in-depth.

Risebro CA, Vieira JM, Klotz L, Riley PR. Characterisation of the human embryonic and foetal epicardium during heart development. *Development*. 2015;142(21):3630-3636. doi:10.1242/dev.127621

Reviewer #2 (Remarks to the Author):

The authors have adequately addressed my critiques. The additional data further strengthen the paper.

We thank and appreciate the comments from reviewer #2 which have helped us to substantially improve the manuscript.

Reviewer #3 (Remarks to the Author):

The authors significantly revised their manuscript and addressed the vast majority of my comments. The major remaining concern is the lack of precision in defining a PEC. This is critical for reproducibility and precision of this work. It would have been very helpful to have had a version of the manuscript with changes highlighted. Also, the figure numbers in the response document didn't match the figure numbers in the manuscript. Thus it took a lot longer to review this revision than it should have.

We apologize for the error in naming the figure number in the previous response document. We have amended these errors in the new document.

1. The revision of the terminology to “proepicardial-like cells” is an improvement. The use of the Cui dataset to compare the proepicardial cells to epicardial cells in vitro is a good idea. I suggest that in light of this the authors revise Figure 1B to identify the genes that distinguish proepicardium from epicardium, including both proepicardial and epicardial genes in this panel. I agree with the authors that these are not the easiest cell types to distinguish (or to map onto cell types in vitro) but all of the genes they list in Figure 1B are also in epicardial cells. A clearer definition of what sets of markers uniquely identify the PECs from other cell types is crucial for reader understanding of the population used in this paper, and for others to isolate and identify this population in their work.

We thank the reviewer for the comment. We have revised our term to name the cells as pre-epicardium to better describe the generated cell population, in view of limited data and evidence to clearly define between proepicardium and epicardium.

Along these lines, Figure 7C shows a putative model of differentiation but as structured it's misleading. As stated in the next point, it's not conclusive that the PECs directly generate SMCs or fibroblasts. Epicardial cells can also generate these cell types. Also, some of the markers listed as PEC are also in epicardial cells but not listed there. I suggest removing this panel or at least substantially revising it to more accurately represent the roles of PECs AND epicardial cells.

We thank the reviewer for the comment and agree that the summary figure is likely an oversimplification and could do more to confuse vs. clarify. We have removed Figure 7C to avoid confusion as suggested.

2. Figure 3A and B don't demonstrate that the PECs directly transit to stromal cells without going through an epicardial state. However this is a tangential point to the core of the manuscript and not necessary to address further.

We thank the reviewer for the comment.

3. The sarcomere structure in Fig 5J is much easier to see. As a minor point, it would help to have a scale bare in Fig 5A. It's not easy for the reader to convert the provided area to length.

We thank the reviewer for the comment, and we have added a scale bar to Fig 5A.